# Cell neighborhood topology directs rare cell population identification

Botao Fa [1,2,6], Chao Huang [1,2,6], Yixin Ma [1,2], Wei Zhao[3], Yungang Xu [4] & Zhengtao Xiao [1,2,5] ✉

Advances in single-cell and spatial technologies have transformed the dissection of cell composition and tissue architecture in complex biological systems. However, identifying rare cells critical to disease pathogenesis and biological processes remains difficult, as their low abundance often masks them among dominant cell populations. We introduce RareQ, a fast and scalable framework for rare cell detection by evaluating cliquishness of each cell's $k$-nearest neighborhood using single-cell omics data. Extensive benchmarking on diverse simulated and real datasets demonstrates that RareQ exceeds existing methods in accuracy, sensitivity, and efficiency. RareQ also excels at identifying both modality-specific and shared rare cells. Its versatility across various biological contexts enables the discovery of functionally distinct rare cells with unique molecular signatures in physiological and pathological context. RareQ's application to spatial transcriptomics data reveals anatomically distinct and clinically relevant rare cell populations. Together, RareQ offers an efficient approach for rare cell discovery, enhancing insights into tissue organization and disease mechanisms.

Rapid development of single-cell and spatial omics technologies has revolutionized our ability to dissect complex tissues at unprecedented resolution[1–3]. These technologies not only enable the identification of diverse cell types and functional states, but also provide insights into spatial organization, cell–cell interactions, and dynamic transitions within tissue contexts[3]. Despite substantial progress in detecting major cell populations using clustering algorithms, accurately identifying and characterizing rare cell types remains a significant challenge. Rare cell populations—often comprising less than 1% of all profiled cells[4–7]—frequently play outsized roles in physiological and pathological processes[8]. For example, circulating tumor cells (CTCs) and cancer stem cells, though scarce, can initiate metastasis and sustain tumor growth[9–11]. In neurological disorders, rare neural stem cells influence neurotoxic activation of reactive astrocytes[12]. Standard clustering algorithms are biased toward main cell populations and often merge or discard small clusters[13]. Accurate detection of rare cell types demands computational approaches that are robust to technical noise and sensitive to weak signals of rare cells.

In response to these challenges, a number of methods have been proposed to improve rare cell identification in single-cell RNA sequencing (scRNA-seq) data. Clustering-based approaches enhance sensitivity to small populations. For instance, RaceID detects rare cell populations by identifying outliers with significantly deviating gene expression profiles from cluster medoids[14]. GiniClust2 integrates Gini index and Fano factor-based gene selection to identify both rare and common cell types[15,16]. CellSIUS adopts a two-step framework involving initial clustering followed by subclustering using genes of bimodal expression within pre-defined clusters[17]. Other strategies eschew

[1]Department of Biochemistry and Molecular Biology, Institute of Molecular and Translational Medicine, Xi'an Jiaotong University Health Science Center, Xi'an, Shaanxi, China. [2]Key Laboratory of Environment and Genes Related to Diseases, Ministry of Education, Xi'an Jiaotong University, Xi'an, Shaanxi, China. [3]Department of General Surgery, the First Affiliated Hospital of Xi'an Jiaotong University, Xi'an, Shaanxi, China. [4]Department of Cell Biology and Genetics, School of Basic Medical Sciences, Xi'an Jiaotong University Health Science Center, Xi'an, Shaanxi, China. [5]Department of Medical Oncology, the Second Affiliated Hospital of Xi'an Jiaotong University, Xi'an, Shaanxi, China. [6]These authors contributed equally: Botao Fa, Chao Huang. ✉e-mail: zhengtao.xiao@xjtu.edu.cn

clustering by treating rare cells as statistical outliers. FiRE assigns each cell a "rareness" score based on its local density using a sketching-based hashing scheme to efficiently identify sparsely populated regions[18]. Similarly, EDGE leverages ensemble learning and dimensionality reduction to highlight consistent outliers across multiple low-dimensional embeddings[19]. As datasets scale to millions of cells, computationally-efficient algorithms are introduced, such as Gap-Clust, a lightweight approach tailored for ultra-large scRNA-seq datasets, balancing speed and accuracy[20]. Meanwhile, scCAD recursively decomposes large clusters based on internal heterogeneity, enabling the recovery of rare subpopulations masked in initial partitions[21]. A critical limitation of current methods is their modality-specific design—most are exclusively optimized for scRNA-seq and cannot generalize to other single-cell modalities such as scATAC-seq or spatial transcriptomics. Methods like MarsGT aim to overcome this by integrating RNA and ATAC modalities through a heterogeneous graph transformer[22]. While promising, their reliance on paired multi-omics datasets limits applicability in practical settings where such data are unavailable. A promising yet underexplored direction for rare cell discovery lies in leveraging the topological properties of low-dimensional data embeddings.

Here, we introduce RareQ, a topology-aware framework that exploits a cell's local neighborhood connectivity to sensitively and efficiently identify rare cell populations. We benchmark RareQ by comparing with existing methods, across a wide range of simulated and real datasets, spanning diverse biological contexts, technological platforms, and multi-modal data types. Its lightweight design makes it computationally efficient and suitable for ultra-large datasets, where many existing methods fail due to memory or runtime constraints. We also demonstrate that RareQ can integrate with multi-omics frameworks to detect rare cell populations that might be missed when analyzing individual regulatory layers separately. Finally, RareQ proves its value in five applications. With airway epithelial scRNA-seq data, it identifies rare cell populations involved in cell cycle regulation, ciliogenesis, and response to interferon functions. In B lymphoma multiome data, RareQ identifies rare myeloid cells such as conventional dendritic cells (cDCs) with enhanced *PD-L1* expression. In Alzheimer's disease (AD) data, RareQ finds microglia and astrocyte populations that reside in reactive states. In spatial transcriptomic data from mouse brain, RareQ uncovers small, anatomically significant subpopulations that were overlooked by previous analyses and existing rare cell detection methods. In human renal cell carcinoma (RCC) spatial data, RareQ captured the tertiary lymphoid structures (TLS) and rare myeloid and proliferative cell populations that were missed by existing algorithms. Taken together, these benchmarking results and biological applications highlight the power and generalizability of RareQ in uncovering rare but biologically significant cell populations from increasingly large and complex single-cell and spatial datasets.

## Results
### Overview of RareQ
The topological structures of single-cell data are typically represented as a low-dimensional *k*-nearest neighbor (kNN) graph which models cells as nodes and proximity relationships as edges. In this network, rare cells tend to form small cliques: their neighbors are tightly interconnected, with strong internal links and few connections to external cells, in contrast to abundant populations. To quantify this property, we define a metric termed neighborhood connectivity ($Q$). $Q$ measures the proportion of internal connections within a cell's neighborhood relative to the total number of connections involving that cell and its neighbors. Conceptually, $Q$ is similar to the clustering coefficient of "small-world" network for measuring the cliquishness of a friendship circle[23], but here it is reformulated as a cell-specific topological measure that characterizes each cell's local neighborhood network. Accordingly, high-$Q$ cells and their tightly knit neighborhoods can

serve as topological anchors around which homologous cells can be preferentially aggregated (Fig. 1a). This metric can be extended to the cluster-level, denoted as $Q_c$, to evaluate cluster independence by calculating the ratio of intra-cluster connections to the total number of connections.

As expected, simulated data analysis shows that rare cells tend to exhibit higher $Q$ values compared to cells from major populations, indicating that rare cells are more likely to form topologically distinct, internally cohesive structures (Fig. 1b). At the population level, both rare and major clusters tend to have $Q_c$ values approaching 1, reflecting their strong internal connectivity. These observations suggest that the $Q$ metric of individual cells enables effective differentiation of rare but biologically relevant clusters, while its cluster-level measurement guarantees the quality of identified clusters. Moreover, its ability to capture diverse topological structures makes it particularly suitable for analyzing spatial transcriptomics data with complex spatial architectures.

Based on the topological property of rare cell populations, we develop a $Q$-guided label propagation strategy to infer both rare and major cell populations (Fig. 1c). The algorithm begins by assigning each cell to its own initial cluster. During iterative propagation, each cell adopts the cluster label of the neighboring cell with the highest $Q$ value if its own $Q$ is lower. In this way, high-$Q$ rare cell cliques serve as stable topological anchors that guide label diffusion outward across the kNN graph until convergence. To minimize misclassification, we incorporate a kNN-voting mechanism that ensures cells connected to multiple clusters are preferentially assigned to the one with greater connectivity. Clusters with average $Q$ values exceeding a predefined threshold are retained as high-confidence rare cell populations. For clusters with lower average $Q$ values, we apply a recursive merging strategy to identify major populations: if the merged cluster exhibits a higher $Q_c$ value than its constituent parts, the merge is accepted; otherwise, the clusters remain separated (see Methods). We also performed ablation studies to isolate RareQ's core components and evaluate their individual contributions to rare-cell detection and overall clustering accuracy (Supplementary Note 1). The results highlight $Q$-guided network propagation and post hoc merging as the central mechanisms underlying RareQ, supporting its robust ability to identify rare cell populations across multiple data modalities and tissue architectures.

### Benchmarking RareQ in both simulated and real scRNA-seq datasets
To assess the performance of RareQ, we compared it against seven existing rare cell detection methods (i.e., scCAD, CellSIUS, RaceID, FiRE, GiniClust2, EDGE, and GapClust) using three simulated experiments encompassing 150 simulated scRNA-seq datasets. These datasets were generated by down-sampling heterogeneous peripheral blood mononuclear cell (PBMC) data, each comprising 5000 cells and representing 5 to 15 distinct cell types (Supplementary Table 1). The $F_1$ score, which effectively captures the trade-off between precision and recall, was used to evaluate the performance of different methods. Compared to other methods, RareQ consistently surpassed them in detecting simulated rare cell populations (Fig. 2a). In Sim-PBMC-1, RareQ outperformed the second-ranked method, scCAD, achieving a 3% improvement; In Sim-PBMC-2, RareQ achieved a 7% improvement compared to the second-ranked method CellSIUS; In Sim-PBMC-3, RareQ achieved 2%, 4%, 6%, and 13% improvements compared to the second-best-performing methods in identifying R1, R2, R4 and R5 rare cell types, respectively.

We further evaluated RareQ on the RNA modality of four human PBMC datasets (PBMC-bench-1, 2, 3, and 4) with known cell type labels (Fig. 2b and Supplementary Fig. 1). We defined rare cell clusters as those comprising less than 1% of the total cell population. RareQ successfully identified 15 out of 17 rare clusters across the four datasets, achieving the highest $F_1$ scores among all methods (Fig. 2c). In contrast,

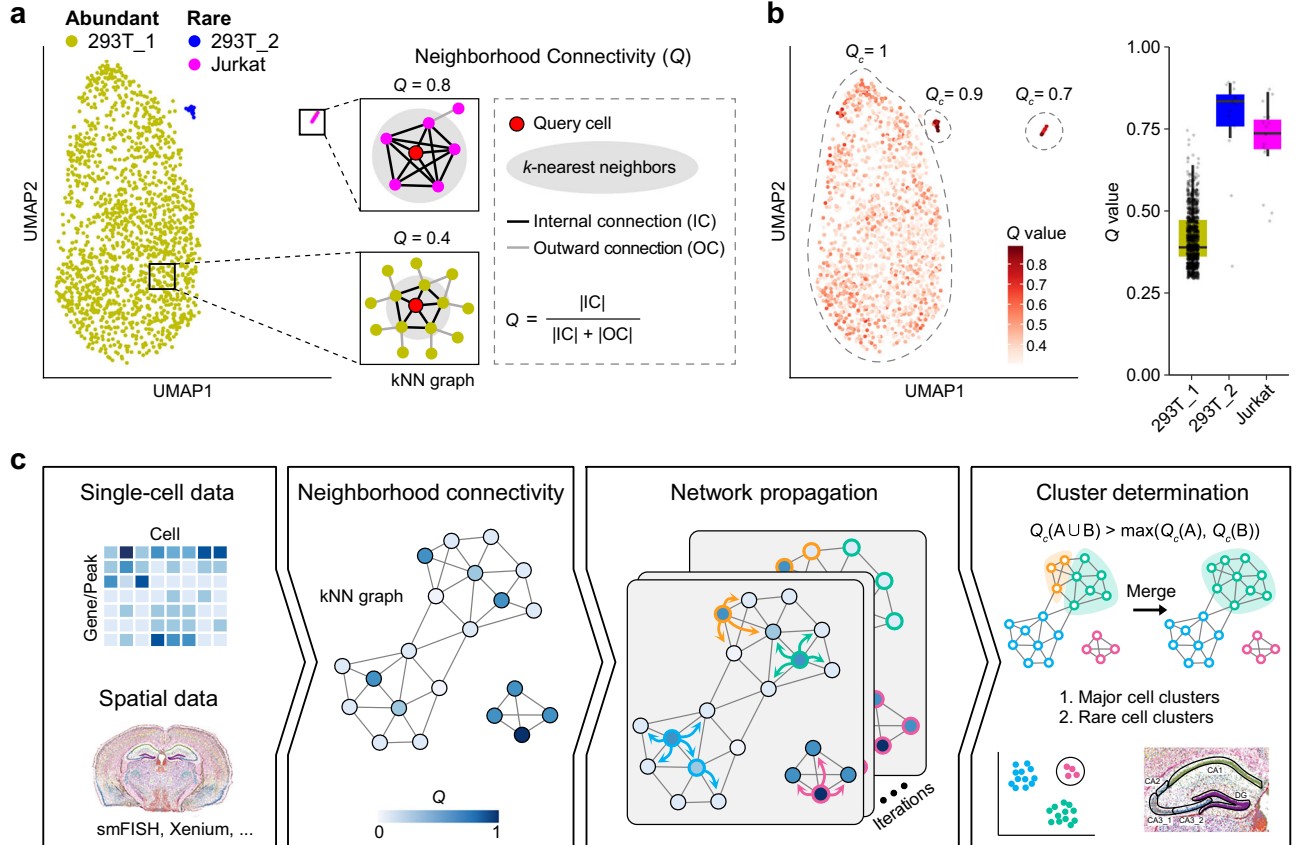

**Fig. 1 | Overview of RareQ framework. a** Schematic illustrating the definition of neighborhood connectivity ($Q$) for a cell, where the rare cells' neighbors tend to be highly interconnected and exhibit higher $Q$ values than the abundant cells. **b** UMAP embeddings colored by the $Q$ value of each cell in the simulated Jurkat dataset (left); Comparison of $Q$ values between rare 293T_2 ($n = 14$), Jurkat ($n = 16$) and abundant 293T_1 cells ($n = 1526$) in their local neighborhoods (right). Boxes extend from the first to the third quartile (Q1 – Q3) with a line in the middle denoting the median. Whisker lines extending from both ends of the box indicate variability outside Q1 and Q3, whose minimum/maximum values are calculated as Q1 − 1.5 × IQR and Q3 + 1.5 × IQR (interquartile range, calculated as Q3 − Q1). **c** Workflow of RareQ in inferring major and rare cell types. RareQ first uses single-cell or spatial omics data as input to construct a $k$-nearest-neighbor (kNN) graph and computes the $Q$ value for each cell; then it propagates the labels of cells whose $Q$ values reach local maxima in kNN graph and generates initial clusters; finally it merges low-quality clusters to generate refined cell clusters. Source data are provided as a Source Data file.

the next-ranked methods, scCAD and RaceID, found only 3 and 6 rare cell clusters, respectively. To confirm RareQ's robust performance, we tested it on 20 additional independent scRNA-seq datasets from various tissues and profiling platforms (Supplementary Table 2). These comparisons consistently showed that RareQ maintained the highest $F_1$ scores (median = 0.91), precision (median = 0.93), and recall (median = 0.91), significantly outperforming other approaches (Fig. 2d and Supplementary Fig. 2). Notably, RareQ was able to identify most rare cell types that other methods missed. RareQ identified 110 of 133 rare cell types, significantly outperforming the second-ranked approach, scCAD, which found only 53 rare cell clusters (Fig. 2e). Across various datasets, UMAP visualizations clearly demonstrated that rare cells identified by RareQ consistently formed distinct clusters, separate from major cell populations (Supplementary Figs. 3–21). Furthermore, we evaluated the global clustering performance of RareQ in identifying both major and rare cell populations using the normalized mutual information (NMI) metric. RareQ consistently outperformed all competing methods across simulated datasets. Notably, it achieved NMI improvements of approximately 17%, 7%, and 4% over the second-best method, CellSIUS, in three simulation experiments (Supplementary Fig. 22). Additionally, we compared RareQ with competing methods across a wide range of parameter combinations and clustering resolutions. Across all settings, RareQ remained robust to parameter choices and achieved superior performance relative to other methods (Supplementary Notes 2–7).

To evaluate RareQ's sensitivity and robustness to the number of differentially expressed genes (DEGs), we conducted experiments using an artificial scRNA-seq dataset and a Jurkat scRNA-seq dataset, each of which comprised two cell types with the minor cell type representing approximately 1% of the total population. Genes that were differentially up-regulated in the rare cell group served as DEGs (211 for the simulated dataset, 148 for the Jurkat dataset, see Methods). In each simulation, an equivalent number of non-DEGs was replaced with randomly selected pre-identified DEGs. This process was repeated 10 times for each number of DEGs, where the average $F_1$ score was computed across iterations for comparison among algorithms. It showed that RareQ precisely identified rare cell types, outperforming its competitors in both datasets (Supplementary Fig. 23). While RaceID performed slightly better or comparably in the artificial scRNA-seq dataset, its performance significantly declined in the Jurkat dataset. Furthermore, using the aforementioned datasets, we confirmed that RareQ is highly stable, showing only minor performance fluctuations near the boundaries of the tested parameter ranges (Supplementary Note 2, Supplementary Figs. 24–28). Batch-only data analysis demonstrated that applying batch correction is critical to prevent spurious "rare populations" that are driven primarily by batch composition rather than biology (Supplementary Note 6). Benchmarking on negative control data indicated that RareQ does not tend to invent rare clusters in relatively homogeneous data when

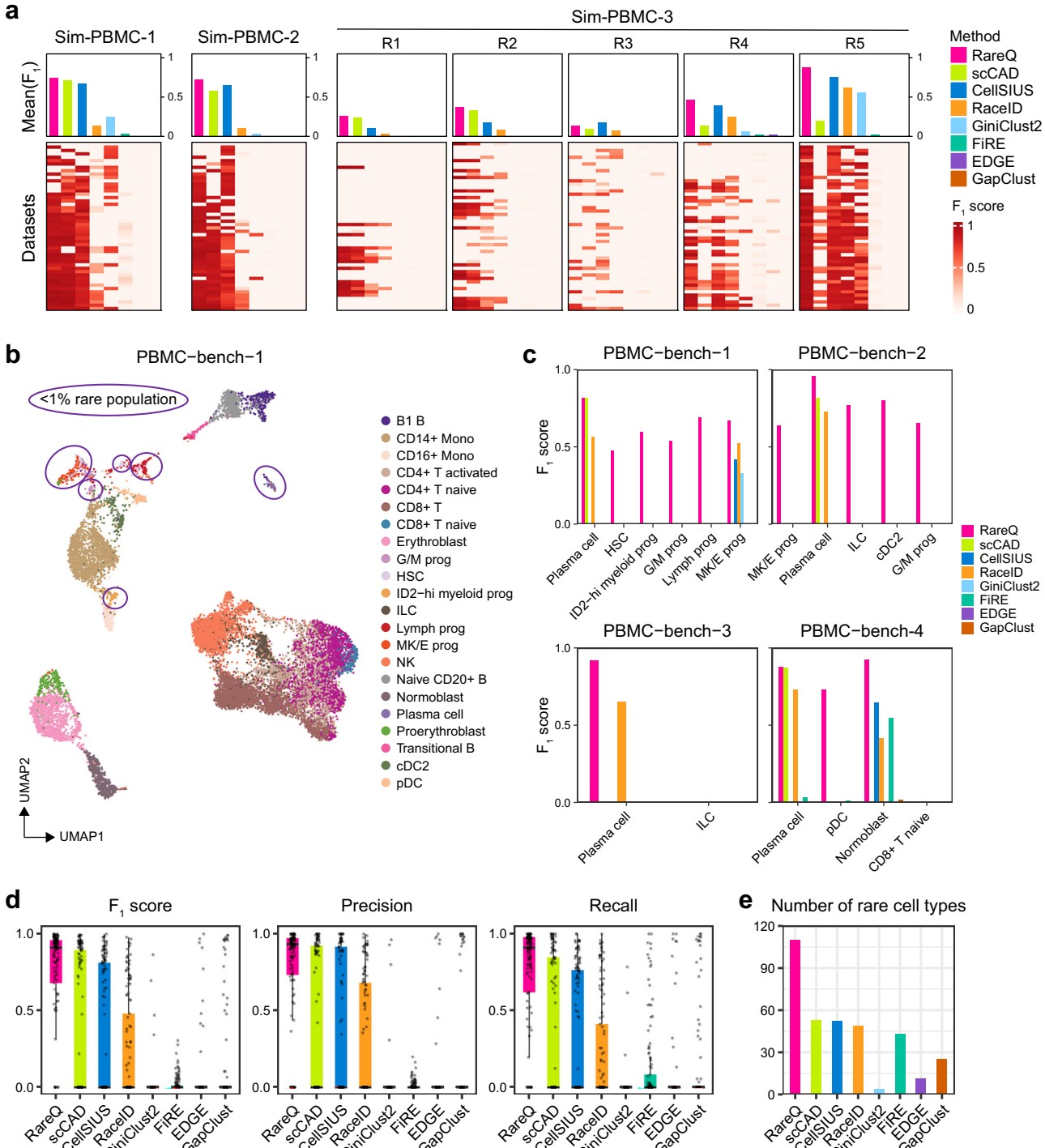

**Fig. 2 | Benchmarking of RareQ for rare cell identification. a** Benchmarking rare cell population identification on Sim-PBMC 1, 2, 3 datasets against existing methods in terms of $F_1$ score. The rows of $F_1$ score heatmaps signify the dataset, while the columns correspond to the specified method. The bar plots above denote the average $F_1$ scores across datasets by specified methods. One rare cell population (R1) in each Sim-PBMC 1/2 dataset, five (R1-R5) in each Sim-PBMC 3 dataset. **b** A UMAP visualizing the rare cell types (<1% total cells, highlighted in purple circles) in the PBMC-bench-1 dataset. **c** Comparative results of RareQ against existing methods on the four human PBMC scRNA-seq datasets (PBMC-bench-1, 2, 3, and 4) via $F_1$ scores. **d** Comparing the distribution of $F_1$ scores, precision, and recall in predicting all rare cell types (<1% total cells) in the 20 real scRNA-seq datasets by specified methods. Boxes extend from the first to the third quartile (Q1 – Q3) with a line in the middle denoting the median. Whisker lines extending from both ends of the box indicate variability outside Q1 and Q3, whose minimum/maximum values are calculated as Q1 − 1.5 × IQR and Q3 + 1.5 × IQR. **e** Comparison of the total number of rare cell types identified by specified methods across the 20 datasets. Source data are provided as a Source Data file.

true rare populations are absent (Supplementary Note 7). Collectively, these findings highlighted RareQ's robustness and its capacity for precise rare cell type identification, even when differential expression signals among cell types were weak.

## RareQ is fast and highly scalable
To evaluate the scalability of RareQ on large datasets, we applied it to a mouse brain scRNA-seq dataset containing 1.47 million cells spanning 31 distinct cell types[24]. Among them, 15 cell types representing less

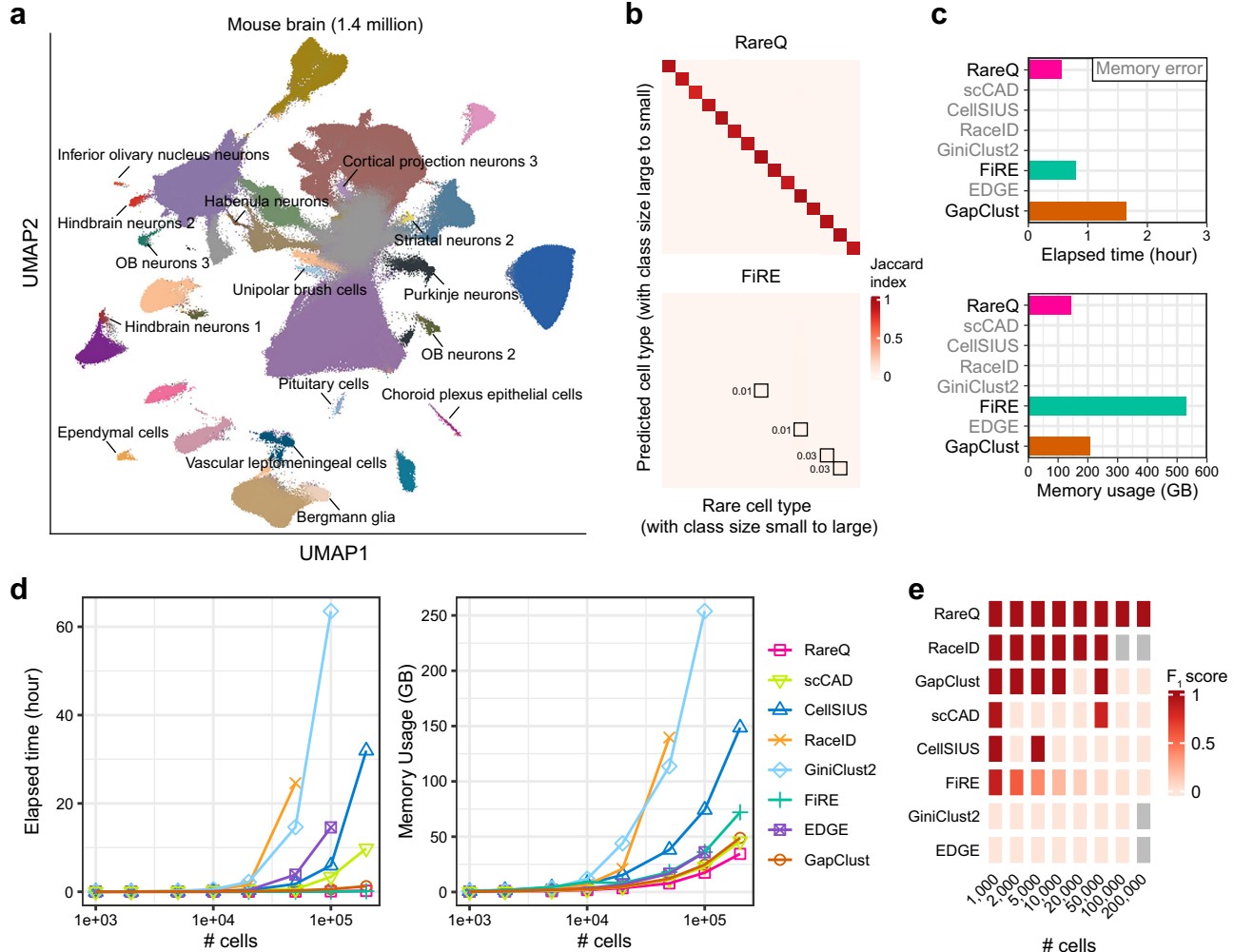

**Fig. 3 | Scalability assessment of RareQ. a** A UMAP visualizing the rare cell types (<1% total cells, highlighted) in the mouse brain dataset comprising ~1.4 million cells. **b** Confusion matrices evaluating the accuracy of specified methods in identifying the fifteen rare cell types via Jaccard index, with a higher value implying higher accuracy. Rows correspond to the prediction, while columns represent the ground truth rare cell types. **c** Time and memory usage of specified approaches in processing the mouse brain dataset. Only RareQ, FiRE and GapClust accomplished the analysis, while the remaining methods failed due to memory error. **d** Time and memory usage of specified methods in processing simulation datasets at different scales. RaceID, GiniClust2 and EDGE failed in large datasets due to memory error. **e** Comparative results of RareQ in identifying the rare cell type with only 10 cells against existing methods on the eight simulation datasets in terms of $F_1$ score. Source data are provided as a Source Data file.

than 1% of the total population were defined as rare cell populations (Fig. 3a). Of the eight tested methods, only RareQ, FiRE, and GapClust successfully processed the full dataset on a computer platform x86_64 architecture, 64 CPUs (Intel Xeon Gold 6226 R, 2.90 GHz) and ~500 GB of memory; the others failed due to memory error. RareQ successfully identified all rare cell clusters, achieving superior accuracy compared to FiRE, whereas GapClust failed to detect these rare cells (Fig. 3b). Notably, RareQ exhibited the highest computational efficiency, requiring the least memory (144 GB) and the shortest runtime (30 min), outperforming FiRE (529 GB, 0.79 h) and GapClust (206 GB, 1.6 h) (Fig. 3c).

We further evaluated RareQ using a series of eight simulated scRNA-seq datasets ranging from 1000 to 200,000 cells, each consisting of one rare cell cluster (10 cells) and one major population. RareQ processed 10,000 cells in just 27 s and scaled to 200,000 cells in 13 min, achieving performance comparable to FiRE (13 s and 7 min) while significantly outperforming GapClust (180 s and 75 min) (Fig. 3d, Supplementary Table 3). Other methods exhibited limited scalability: RaceID failed on datasets with over 50,000 cells; GiniClust2 and EDGE consumed over 253 GB and 35 GB of memory in processing 100,000

cells, respectively, while both crashed on datasets exceeding 100,000 cells (Supplementary Table 4); When processing 200,000 cells, scCAD and CellSIUS required over 10 h and 32 h, respectively. In contrast, RareQ was the only method that successfully identified rare cells in datasets exceeding 100,000 cells. Comprehensive comparisons across three evaluation metrics−$F_1$ score, precision, and recall−further confirmed RareQ's high sensitivity and accuracy (Fig. 3e, Supplementary Fig. 29). Taken together, RareQ exhibited exceptional speed and highest memory efficiency on large-scale datasets.

## RareQ enables rare cell detection from multi-modal single-cell data

To evaluate the applicability of RareQ on multi-modal single-cell data, we applied RareQ to a CITE-seq dataset of human bone marrow mononuclear cells, which includes gene expression profiles and measurements of 25 surface proteins across 30,672 cells spanning 27 cell types[25]. We aimed to evaluate RareQ's ability to detect rare cell populations−which might be missed in single-modality analyses−by integrating it with the weighted nearest neighbor (WNN) framework, a recently developed approach for combining multiple modalities

measured within the same cell[25]. WNN assigned cell-specific weights to each modality according to its relative information content and generated a unified low-dimensional embedding that retains modality-specific signals (Supplementary Fig. 30a). Notably, several T cell subtypes were indiscernible in the RNA modality but clearly resolved in the protein modality. For example, CD8⁺ naive T cells, which showed high transcriptional similarity to CD4⁺ T cells, were clearly separated in embedding space of the protein profiles.

Using curated cell type annotations as ground truth, we benchmarked RareQ across three settings: gene expression alone, protein expression alone, and WNN-integrated embeddings. RareQ successfully identified 7 out of 9 rare cell types across all scenarios; however, some of them were modality-specific (Supplementary Fig. 30b). Specifically, progenitor megakaryocytes (prog_MK) and hematopoietic stem cells (HSCs) were detected in both gene expression data and WNN-integrated data, while CD56⁺ bright NK cells and regulatory T cells (Tregs) were exclusively identified in the protein modality (Supplementary Fig. 30c). Compared to existing algorithms, RareQ consistently achieved the highest $F_1$ scores across all rare cell types, with the sole exception of CD56⁺ bright NK cells, which only RaceID detected (Supplementary Fig. 30d). Collectively, these results demonstrated that RareQ can be effectively applied to WNN-integrated multi-modal data to enhance rare cell identification. However, some rare cell types were detectable only in individual modalities, underscoring the importance of performing independent analyses prior to integration to maximize detection sensitivity.

## RareQ outperforms the multi-omics integration method MarsGT in rare cell inference

Recent efforts, such as MarsGT, have introduced deep learning frameworks designed to use both scRNA-seq and scATAC-seq data to identify rare cell populations[22]. However, integrating data modalities that have different dimensionalities and distributions can lead to alignment errors or overlooked modality-specific signals. In contrast, RareQ flexibly supports both independent analysis of each modality and integrated analysis in combination with the WNN framework. First, we benchmarked RareQ against MarsGT using three simulated experiments of PBMC datasets (Sim-PBMC 1–3), each comprising 50 paired scRNA-seq and scATAC-seq profiles. With 150 simulation runs, RareQ's $F_1$ scores were consistently and significantly higher than MarsGT's, especially when utilizing WNN-integrated embeddings (Fig. 4a).

Next, we applied RareQ and MarsGT to the aforementioned four PBMC multi-omics datasets (PBMC-bench 1–4) with paired RNA and ATAC data, each containing 2 to 6 rare cell types (<1% total population). In PBMC-bench-1 and -2, RareQ accurately identified all rare cell types using both the RNA modality and the WNN-integrated embedding, whereas MarsGT detected only one rare cluster in each dataset—MK/E prog in PBMC-bench-1 and plasma cells in PBMC-bench-2 (Fig. 4b, Supplementary Figs. 31, 32). In PBMC-bench-3 and -4, MarsGT failed to detect any rare cell types, while RareQ successfully identified one of the two rare clusters in PBMC-bench-3 and all rare cell types in PBMC-bench-4 (Fig. 4b, Supplementary Figs. 33, 34).

In addition to the annotated rare cell types, RareQ effectively discovered novel rare subpopulations that possessed significant chromatin features distinct from canonical lineages. An example from PBMC-bench-4 showed RareQ identified two granulocyte/macrophage progenitor (G/M prog) clusters, two plasmacytoid dendritic cell (pDC) clusters, and two lymphoid progenitor (Lymph prog) cell clusters, none of which were detected by MarsGT (Fig. 4c). Differential analysis of gene expression and chromatin accessibility confirmed the distinctiveness of these subtypes, with particularly pronounced differences in their chromatin accessibility profiles (Fig. 4d, Supplementary Fig. 35). Notably, in terms of computational efficiency, we observed that the GPU-accelerated version of MarsGT completed the analysis in

more than 10 h (10 epochs for training) on the computer with 2 GPUs of NVIDIA RTX A6000 with 48 GB memory, while the CPU version required seven days when using 128 CPUs (AMD EPYC 7543 32-Core Processor) to process the PBMC-bench-2 dataset (-15,000 cells). In contrast, RareQ completed the same analysis in just one minute. RareQ demonstrated consistently superior performance to MarsGT across ten independent paired multi-modal datasets, even when MarsGT was run with its optimal hyperparameters and with WNN-derived embeddings (Fig. 4e, Supplementary Figs. 36–46). This strong performance was also reflected in its robust global clustering capabilities confirmed across a comprehensive collection of diverse datasets (Fig. 4f, Supplementary Fig. 47). These results collectively highlight that RareQ is capable of uncovering novel, biologically meaningful cell clusters.

## RareQ enables comprehensive identification of rare airway epithelial cell types

While basal, secretory, and ciliated cells are the most common airway epithelial cells, rare epithelial subtypes are increasingly recognized for their ability to sense environmental stimuli and initiate repair[26,27]. Their dysregulation often contributes to disease development. To comprehensively identify rare cell populations within the airway epithelium, we applied RareQ alongside competing methods to a scRNA-seq dataset from mouse tracheal epithelium[26] (Fig. 5a). Ten of RareQ's identified clusters, which had relatively small cell populations, exhibited significantly higher $Q$ values, allowing them to be clearly distinguished from more abundant cell types (Fig. 5b). Among these, six of RareQ's identified clusters showed strong agreement with the original study's annotations as indicated by high Jaccard similarity (Fig. 5c). Furthermore, the top five upregulated genes in each RareQ-identified rare cluster matched the marker features of their corresponding annotated cell types (Fig. 5d). Note that other competing methods failed to fully recover these rare populations. For example, RaceID—the second-best-performing method—was unable to detect the Krt4/13⁺ cell population (Fig. 5a).

In addition to the known rare cell clusters, RareQ also identified four novel rare clusters (C3, C4, C5, and C10) that diverged from the original annotations. For example, cluster C3 showed decreased expression of histone genes such as *Hist1h2ap* relative to canonical cycling basal cells (C6). Cluster C4 exhibited markedly elevated expression of genes like *Ccdc67* and *Ccno* compared to abundant cell cluster C11 (Fig. 5e). C4 also strongly expressed genes related to chromosome condensation, spindle midzone assembly, and intraciliary transport (Supplementary Fig. 48). Clusters C5 and C10, which resembled secretory clusters C12 and C13, were marked by strong upregulation of interferon-stimulated genes such as *Ifit3*, *Ifit3b*, and *Isg15*, as well as *Foxj1*, *Lrrc23*, and *Tuba1a* (Fig. 5e). Pathway-level analysis revealed that C5 upregulated genes associated with interferon-α and interferon-β response pathways, while C10 showed higher enrichment for pathways associated with intraciliary transport, axoneme assembly, and early endosome to Golgi transport (Supplementary Fig. 48). In addition, we confirmed the presence of C3, C4, and C10 in specific regions of the airway lumen across three independent human spatial transcriptomics datasets[28,29], using the corresponding RareQ-derived marker genes (Supplementary Fig. 49). Together, these results demonstrated that RareQ enables comprehensive identification of rare airway epithelial cells and uncovers novel functional cell clusters.

## RareQ identifies *PD-L1* high dendritic cell subsets in B cell lymphoma

To evaluate the applicability of RareQ in cancer-related contexts, we analyzed 14,566 paired scRNA-seq and scATAC-seq profiles obtained from intra-abdominal lymph node tumor tissue of a patient diagnosed with diffuse small lymphocytic lymphoma. RareQ not only recapitulated the three rare populations (17, 18, and 16) previously detected by

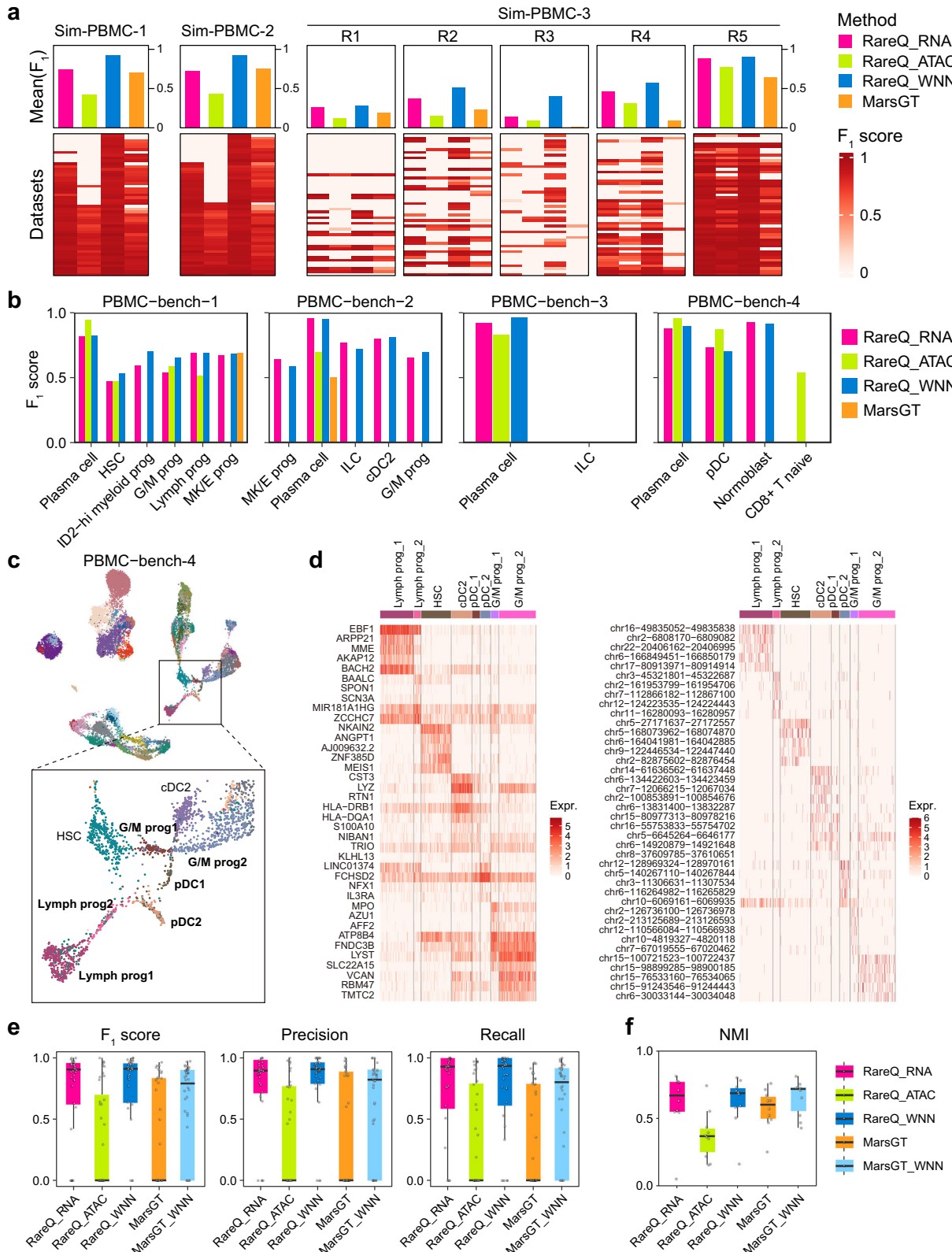

MarsGT, i.e., B (13), CD4 T (11), and CD4 T (12) cells, but also discovered additional rare populations, particularly within the myeloid lineage (Fig. 6a). These included clusters 5, 7, 9, 12, 15, and 23, which displayed distinct molecular signatures (Fig. 6b).

Annotation with canonical marker genes classified clusters 9 and 23 as macrophages, clusters 5 and 7 as cDCs, and clusters 12 and 15 as monocytes (Fig. 6c). Clusters 5 and 7 exhibited pronounced antigen-

presenting features, whereas clusters 9 and 23 were enriched for macrophage activation signals (Fig. 6d). Importantly, cluster 5 demonstrated high expression of *PD-L1* and activation of the PD-L1/PD-1 checkpoint signaling pathway. Consistently, the transcription factors STAT1 and NFKB1—key regulators of *PD-L1* expression—showed markedly increased expression and chromatin accessibility at their genomic loci (Fig. 6e, f and Supplementary Fig. 50). Given the critical

**Fig. 4 | Benchmarking of RareQ on multi-omics single-cell data. a** Benchmarking RareQ against MarsGT in the paired Sim-PBMC 1, 2, and 3 datasets via F₁ score. RareQ was applied to each modality independently (RareQ_RNA for RNA data, RareQ_ATAC for ATAC data, RareQ_WNN for WNN-integrated data), while MarsGT integrates both modalities. **b** Comparative results of RareQ against MarsGT on the four human PBMC paired scRNA-seq datasets (PBMC-bench-1, 2, 3 and 4) with scATAC-seq data via F₁ scores. **c** Magnified view of Supplementary Fig. 34 showing the progenitor cell types identified by RareQ on ATAC modality. **d** Heatmaps of cell-

type-specific markers of identified cell subsets in expression (left) and accessibility (right). Benchmarking RareQ against MarsGT under optimal parameter settings using 10 paired scRNA-seq and scATAC-seq datasets on (**e**) rare cell detection via F₁ score, precision and recall, and (**f**) global clustering via NMI. Boxes extend from the first to the third quartile (Q1 – Q3) with a line in the middle denoting the median. Whisker lines extending from both ends of the box indicate variability outside Q1 and Q3, whose minimum/maximum values are calculated as Q1 – 1.5 × IQR and Q3 + 1.5 × IQR. Source data are provided as a Source Data file.

role of *PD-L1*–high cDCs in dampening T cell activation and shaping responses to immune checkpoint blockade, precise identification of this population may provide valuable insights for cancer immunotherapy. Additionally, RareQ identified rare populations not detected by MarsGT, including fibroblasts (3), plasma cells (6), endothelial cells (8), and pDCs (11) (Supplementary Fig. 51).

## RareQ uncovers rare microglial and astrocytic subsets in Alzheimer's disease

To assess the utility of RareQ in neurodegenerative disease, we applied it to single-nucleus RNA-sequencing (snRNA-seq) data from prefrontal cortex tissues of four healthy individuals and four patients with Alzheimer's disease[30] (AD). RareQ resolved 25 distinct clusters, including five endothelial groups, seven astrocyte groups, three microglial groups, four oligodendrocyte groups, and six oligodendrocyte precursor cell (OPC) groups (Fig. 7a).

Within the microglial population, rare subsets 15 and 16 displayed distinct transcriptional programs (Fig. 7b). Cluster 15 was defined by elevated expression of inflammatory and phagocytosis-associated genes such as *IFI44L* and *NAMPT*, whereas cluster 16 expressed neuronal connectivity-related genes including *PCDH9* and *IL1RAPL1*. Abundance analysis revealed a strong enrichment of cluster 15 in AD patients (Fig. 7c, Supplementary Table 6). This increase of C15 in AD was further confirmed by its signature gene expression in an independent bulk RNA-seq cohort (Fig. 7d). Functionally, this subset showed enhanced activity in phagocytosis, myeloid leukocyte activation, ion transport, and cytokine-mediated signaling pathways, all of which are linked to the clearance of AD-associated pathological products (Fig. 7e). SCENIC[31] analysis revealed enhanced activity of transcription factors linked to pro-inflammatory regulation, including STAT1, ELF1, ETV6, and FOXP1, as well as context-specific immune and phagocytic regulators such as RUNX2 (Fig. 7f).

Astrocyte heterogeneity was also evident across seven subpopulations (clusters 5, 7, 12, 18, 19, 20, and 24) (Fig. 7g). Clusters 5, 12, and 20 were preferentially enriched in AD patients, with cluster 12 consistently detected across multiple individuals (Fig. 7h). All three expressed markers of reactive astrocytes; however, cluster 12 exhibited a distinct molecular profile (Fig. 7i), suggesting alternative reactivation mechanisms under pathological stress. ssGSEA analysis revealed that cluster 12 was significantly enriched for chaperone-mediated autophagy and amyloid fibril formation pathways (Fig. 7j).

RareQ also identified rare endothelial, oligodendrocyte, and OPC subpopulations, each characterized by distinct molecular signatures (Supplementary Fig. 52). In contrast, scCAD, CellSIUS, RaceID, GiniClust2, FiRE, EDGE, and GapClust failed to detect microglial cluster 15 or astrocyte clusters 5 and 12 (Supplementary Fig. 53).

## RareQ identifies spatial architecture of rare cells from multi-platform spatial transcriptomics data

Advances in in situ hybridization (ISH) and in situ sequencing (ISS) technologies have enabled high-resolution spatial transcriptomics through platforms such as Enhanced ELectric Fluorescence in situ Hybridization (EEL FISH) and Xenium[32,33]. These platforms can map the spatial distribution of hundreds of RNA species at subcellular resolution, allowing for the segmentation and identification of individual

cells based on defined gene panels. We evaluated RareQ's capability to detect rare cell types and resolve fine-grained spatial architectures using spatial omics data. We first applied RareQ to an EEL FISH-derived spatial transcriptomics dataset from a sagittal mouse brain section, comprising 127,591 cells and 440 genes (Supplementary Fig. 54a). Notably, without relying on any external reference data, RareQ identified 13 rare cell clusters (each comprising <1% of total cells) and revealed several small yet anatomically meaningful populations corresponding to the rostral migratory stream (RMS), hippocampus, and the thin Purkinje layer of the cerebellum[32] (Supplementary Fig. 54b). The spatial expression of marker genes such as *Sox11*, *Prox1*, and *Gdf10* supported the anatomical identities of these cell clusters (Supplementary Fig. 54c). The side-by-side comparison between RareQ-derived annotations and the original cluster labels was presented in Supplementary Fig. 54d. By comparison, scCAD, CellSIUS, FiRE, and GapClust either did not detect rare cell clusters or identified clusters that were diffusely distributed without correspondence to anatomical structures (Supplementary Fig. 55).

RareQ also uncovered previously unreported rare-cell populations, exemplified by a choroid plexus epithelial subgroup (C1) marked by high expression of *Foxj1*, *Otx2*, and *Lratd2*, and spatially embedded within the major epithelial cluster (C11) (Supplementary Fig. 56a–c). The spatial positioning of C1, together with its strong expression of cilium-associated genes such as *Foxj1*, suggests that this population corresponds to ciliated epithelial cells of the choroid plexus. Because cilia play essential roles in cerebrospinal fluid flow and maintenance of the ventricular microenvironment, identifying this rare ciliated-like epithelial population provides a higher-resolution cellular framework for dissecting the structural and functional organization of the ventricular system. We confirmed the presence of this cell type in an independent spatial dataset from early mouse brain development[34], which revealed a corresponding epithelial population expressing *Foxj1* and *Otx2* adjacent to the main *Ttr*-defined choroid plexus clusters (Supplementary Fig. 56d, e). Further validation using a high-resolution choroid plexus scRNA-seq dataset demonstrated that epithelial populations could be accurately mapped to their spatial counterparts in the EEL FISH data (Supplementary Fig. 56f, g). Among the 16 epithelial clusters identified by RareQ, several groups—including C3, C4, and C5—exhibited elevated expression of *Foxj1*, *Otx2*, and *Lratd2*, mirroring the molecular signatures of the spatially defined C1 population (Supplementary Fig. 56h). Together, these findings strongly support that the rare populations identified by RareQ within the spatial context represent authentic, specialized functional states rather than technical artifacts.

We next applied RareQ to a second spatial transcriptomics dataset from a mouse brain coronal section, generated using the Xenium platform. Compared to EEL FISH, Xenium provided lower inter-cell-type contamination, with 162,033 cells profiled across 248 genes[33]. RareQ identified 199 cell clusters, of which 170 contained fewer than 1% of total cells, reflecting its sensitivity to rare populations (Fig. 8a). We further confirmed with CHOIR[35]-based testing that these clusters represent statistically distinct populations rather than overclustering artifacts (Supplementary Fig. 57a). Furthermore, we used the silhouette coefficient—an effective internal metric for assessing cluster count appropriateness and overall quality—to independently evaluate

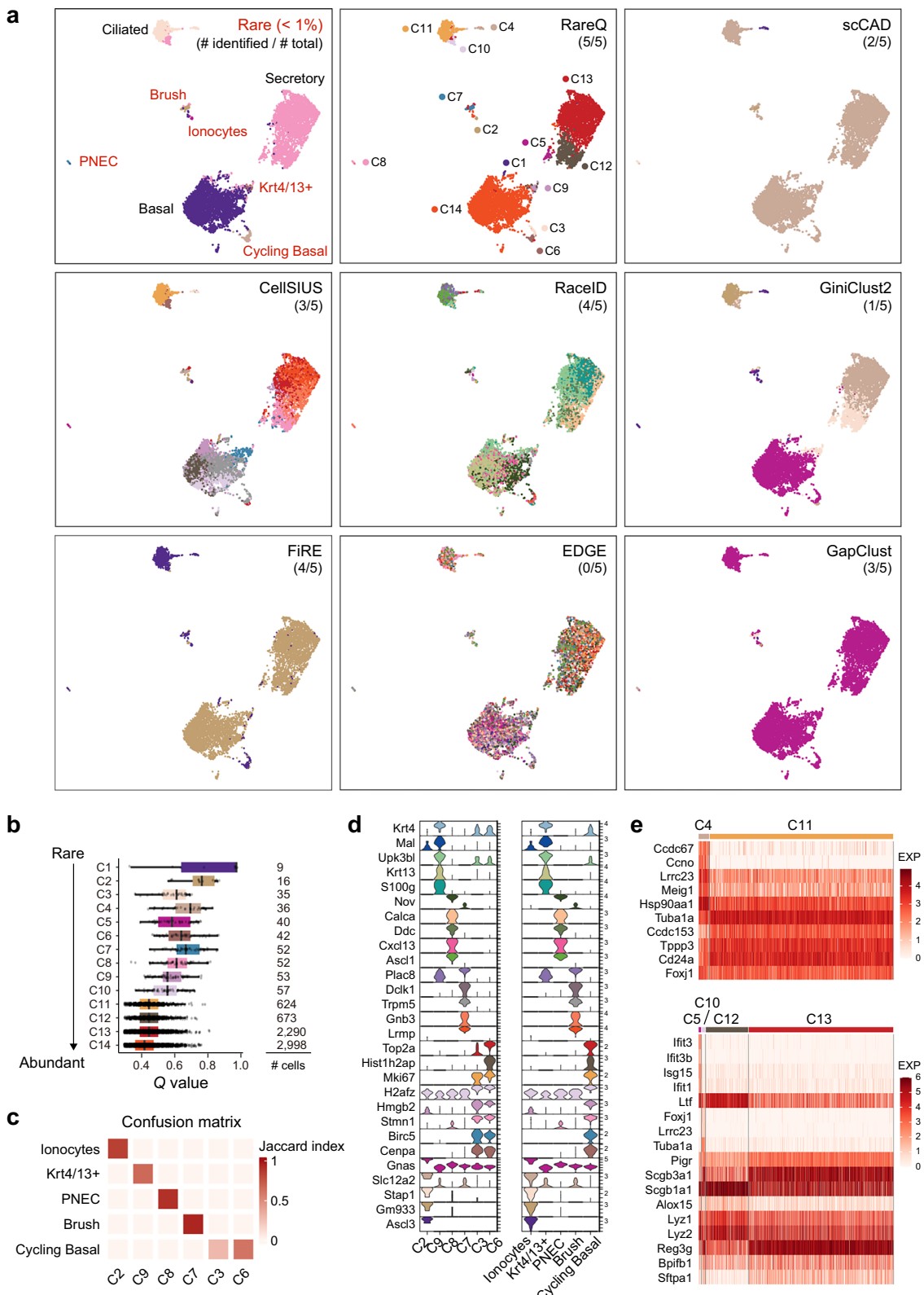

RareQ's clustering quality against Seurat (Supplementary Fig. 57b). It showed that RareQ's clusters generally achieved a higher-quality score than those generated by Seurat. The spatial organization of these clusters revealed molecularly distinct anatomical regions, evidenced by differential gene expression when compared to their most similar major clusters (Supplementary Fig. 58). In particular, RareQ delineated five distinct cell clusters within the hippocampus, corresponding to CA1, CA2, CA3, and dentate gyrus (DG) neurons. Notably, RareQ recovered the CA2 population, a region located between CA1 and CA3 that the original data analysis had not uncovered. This cluster was characterized by the high expression of *Stard5*[36] and *Plcxd3* (Fig. 8b). The CA1 and DG regions were confirmed by their specific markers, including *Wfs1*, *Fibcd1*, *Slc44a5*, and *Plekha2* (Fig. 8c). Interestingly, RareQ notably segmented the CA3 region into two distinct subclusters

**Fig. 5 | RareQ identifies functional rare cell types in airway epithelium. a** The UMAPs of the airway epithelium dataset by Plasschaert et al. colored by ground truth cell types (rare cell populations constituting less than 1% of the total cells are highlighted in red) and predicted cell clusters by specified tools. **b** Comparing the distribution of *Q* values between identified cell clusters by RareQ. Boxes extend from the first to the third quartile (Q1 – Q3) with a line in the middle denoting the median. Whisker lines extending from both ends of the box indicate variability outside Q1 and Q3, whose minimum/maximum values are calculated as Q1 − 1.5 × IQR and Q3 + 1.5 × IQR. **c** Confusion matrix evaluating the accuracy of RareQ in identifying the five rare cell types via Jaccard index, with a higher value implying higher accuracy. Rows represent the ground truth rare cell types, while columns correspond to predicted clusters. **d** Violin plots showing the expression distribution of the most differentially up-regulated genes of each rare cell type in six identified cell clusters and five reported rare cell types for comparison. **e** Heatmaps displaying the expression of top up-regulated genes of newly identified clusters within ciliated (top) and secretory cells (bottom). Source data are provided as a Source Data file.

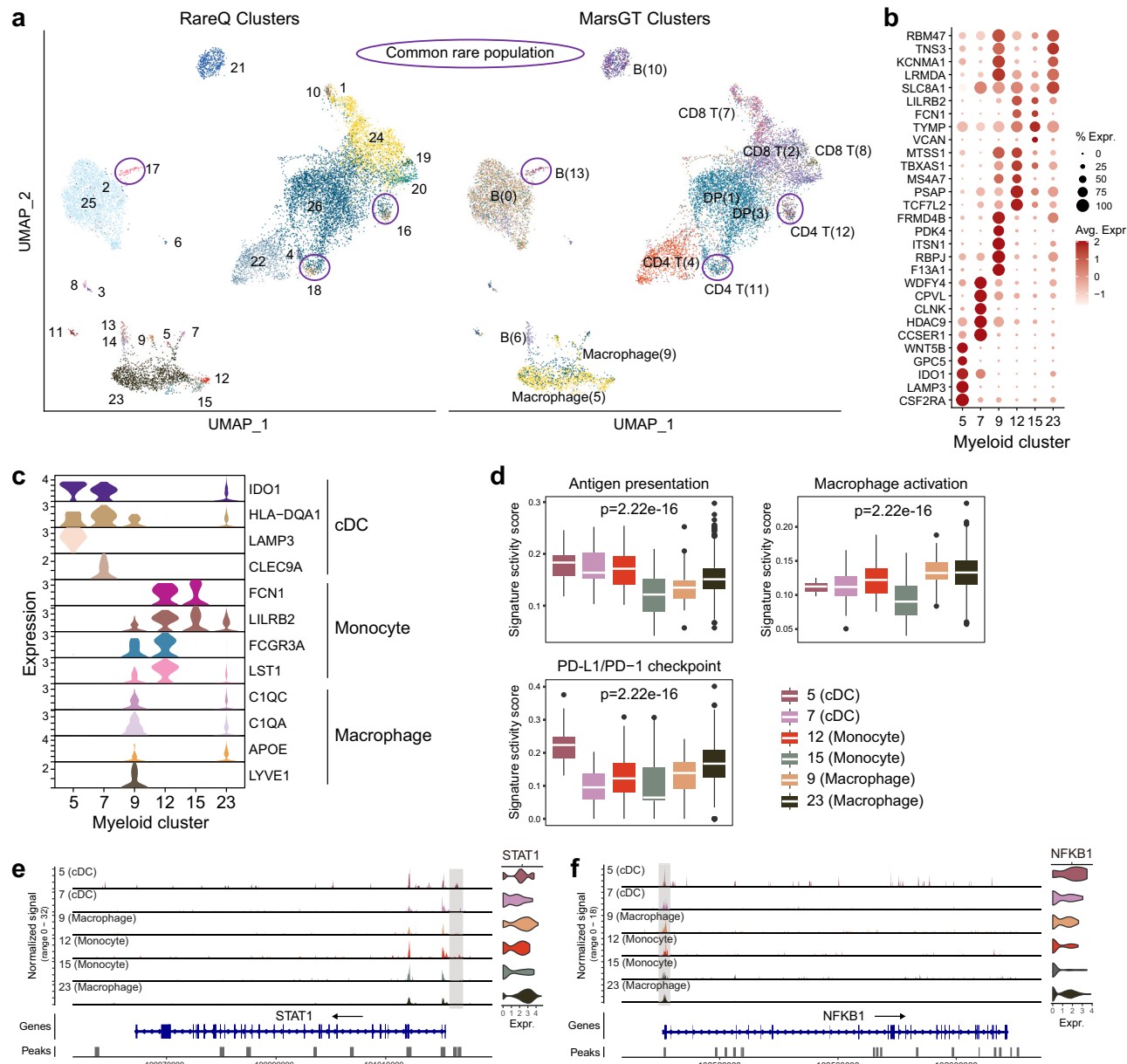

**Fig. 6 | RareQ identifies rare myeloid cell types in B lymphoma. a** UMAPs visualizing cell clusters predicted by RareQ and MarsGT, with the common rare cell types identified by both highlighted. **b** Dot plot demonstrates the expression value and proportion of the top DEGs between the myeloid cell clusters predicted by RareQ. **c** Violin plot shows the expression of marker genes within the myeloid clusters for annotation. **d** Box plots comparing gene signature scores of antigen presentation, macrophage activation, and PD-L1 pathways among myeloid clusters. Boxes extend from the first to the third quartile (Q1 – Q3) with a line in the middle denoting the median. Whisker lines extending from both ends of the box indicate variability outside Q1 and Q3, whose minimum/maximum values are calculated as Q1 − 1.5 × IQR and Q3 + 1.5 × IQR. *p*-values were calculated using ANOVA. Chromatin accessibility profiles for gene (**e**) *STAT1* and (**f**) *NFKB1*. Source data are provided as a Source Data file.

(CA3_1 and CA3_2) that standard clustering algorithms failed to resolve (Supplementary Fig. 59). Differential expression analysis revealed a gradient pattern of gene expression across the two clusters. Specifically, *Strip2* displayed a decreasing trend from CA3_1 to CA3_2, while *Rspo2*, *Nwd2*, and *Plcxd2* showed progressively increasing expression in the same direction (Fig. 8d). These findings highlighted RareQ's ability to uncover spatial heterogeneity that was not previously resolved.

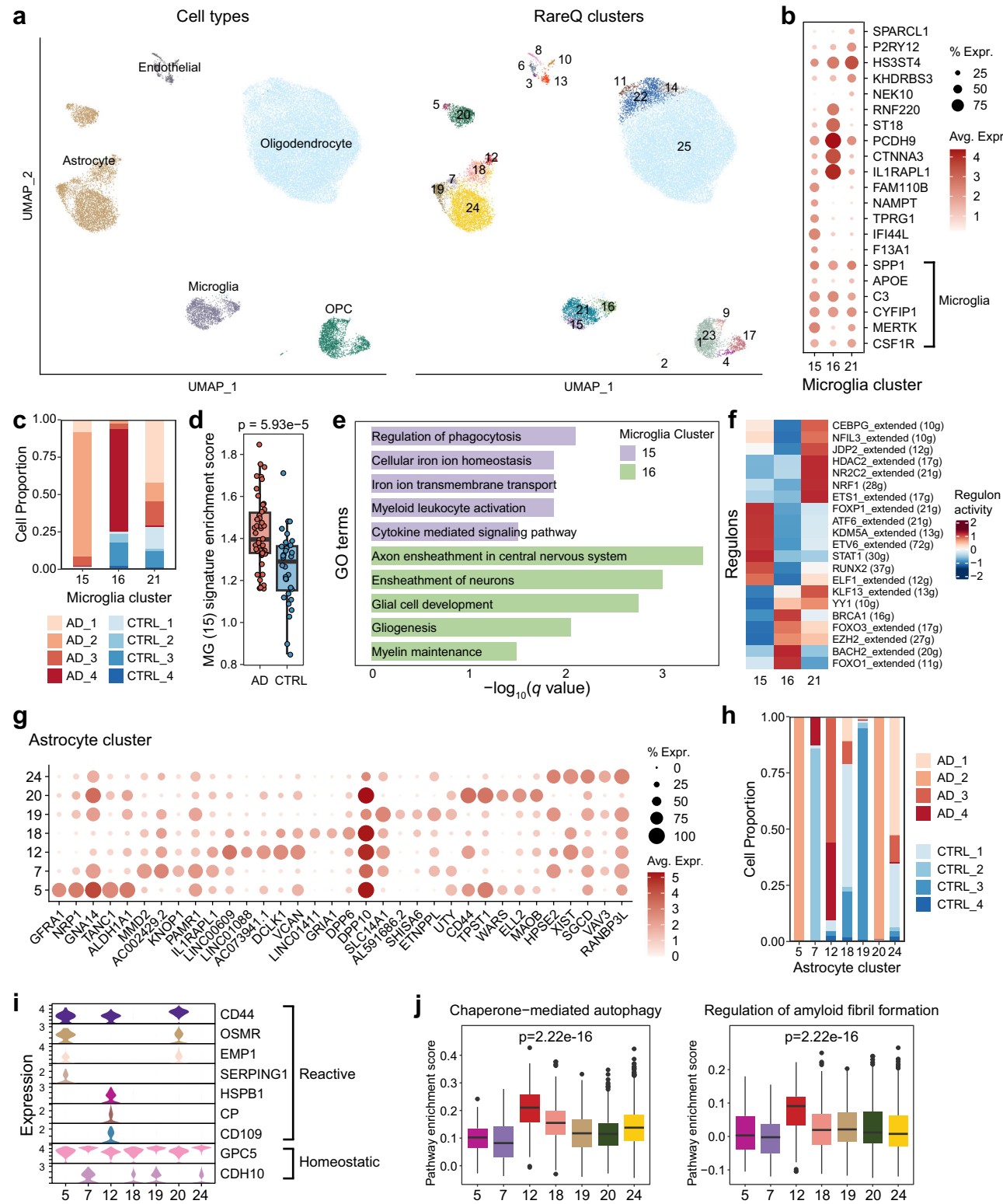

To validate the existence of these two CA3 subpopulations, we further applied RareQ to an independent hippocampal single-cell RNA-sequencing dataset comprising 12,418 cells[37], which were partitioned into 76 clusters (Fig. 8e). The CA2 and CA3 populations were annotated using the established markers *Cpne4*, *Nectin3*, and *Plcxd3* (Fig. 8f). Within the CA3 subclusters, we identified distinct molecular signatures, including *Stard5*, *Strip2*, *Rspo2*, and *Nwd2*, consistent with our spatial transcriptomic observations (Fig. 8g). Comparative transcriptomic analysis revealed significant differences between CA2 and

the three CA3 subclusters (Fig. 8h). ssGSEA-based pathway enrichment analysis further highlighted divergent activities in amine binding, G-protein coupled purinergic receptor (GPCR) signaling, and phosphate ion binding across these subpopulations (Fig. 8i). Moreover, previous evidence from RNA in situ hybridization and RNAscope assays corroborated the molecular features of these two CA3 subtypes[37–40]. Besides, we validated our findings using two independent mouse brain spatial transcriptomics datasets from 10X genomics datasets (Supplementary Fig. 60).

**Fig. 7 | RareQ identifies rare microglia and astrocyte populations in Alzheimer's disease. a** UMAP projections visualizing the major cell types and cell clusters predicted by RareQ. **b** Dot plot illustrating the expression levels and proportions of top DEGs across microglial clusters predicted by RareQ. **c** Bar plot showing the proportions of cells derived from AD patients and healthy control (CTRL) individuals within each microglial cluster. **d** Independent bulk RNA-seq dataset analysis confirms the significant enrichment of C15 microglial population in AD patients ($n = 46$) versus control individuals ($n = 32$), with $p$-value calculated using two-sided Wilcoxon test. **e** Bar plot displaying the enriched pathways of microglial clusters 15 and 16. **f** Heatmap representing the inferred regulon activities across microglial clusters.

**g** Same as (**b**), but for astrocyte clusters. **h** Same as (**c**), but for astrocyte clusters. **i** Violin plot showing the expression of marker genes (reactive and homeostatic) within astrocyte clusters. **j** Box plots comparing gene signature scores for the autophagy and regulation of amyloid fibril formation pathways across astrocyte clusters. The cell numbers in clusters 5, 7, 12, 18, 19, 20, and 24 are 80, 92, 171, 458, 576, 902 and 3595, respectively. $p$-values were calculated using ANOVA. Boxes extend from the first to the third quartile (Q1 − Q3) with a line in the middle denoting the median. Whisker lines extending from both ends of the box indicate variability outside Q1 and Q3, whose minimum/maximum values are calculated as Q1 − 1.5 × IQR and Q3 + 1.5 × IQR. Source data are provided as a Source Data file.

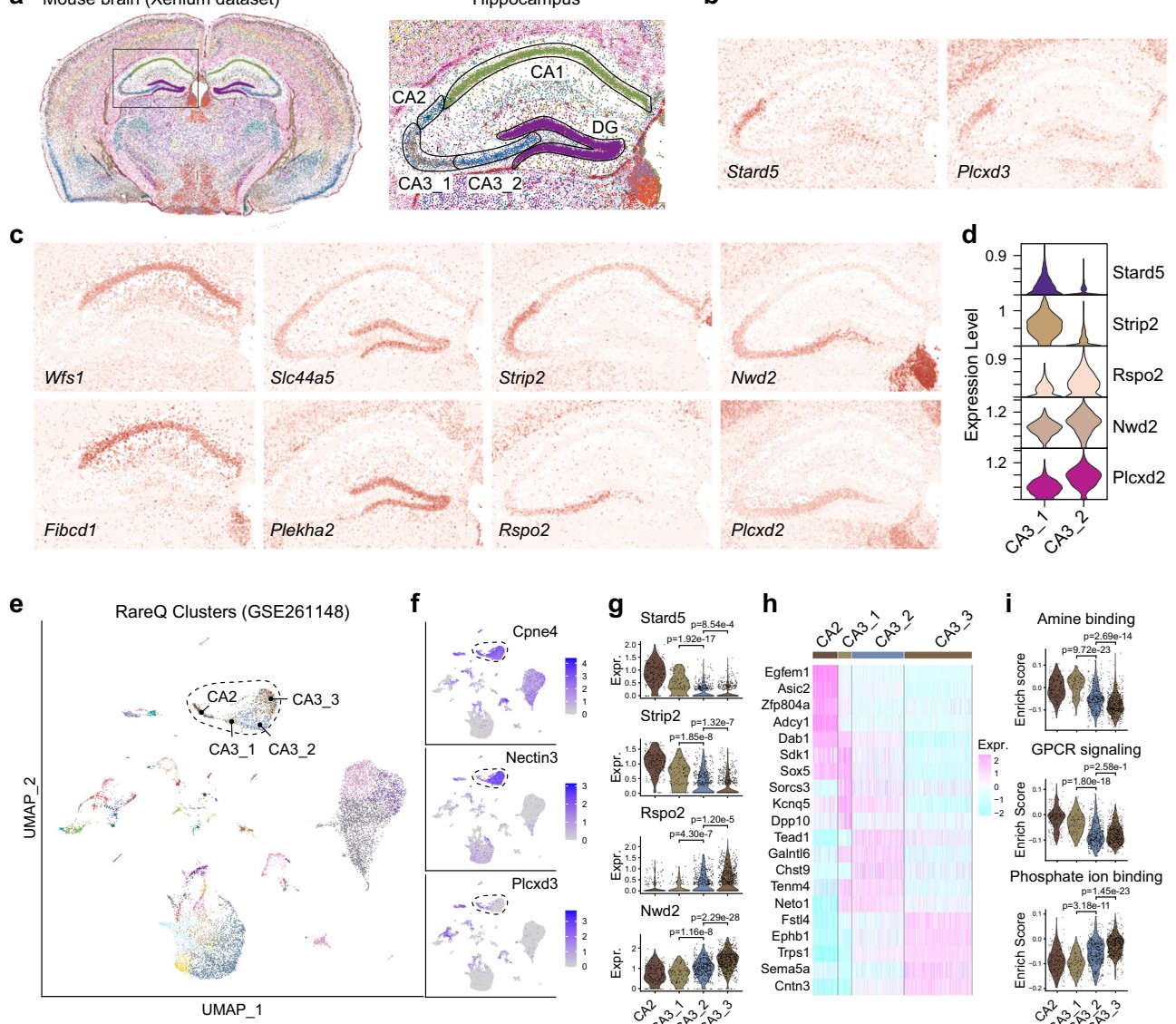

**Fig. 8 | Spatial mapping of rare cells in the mouse brain using RareQ. a** Spatial map of Xenium mouse brain data where every dot is a single cell colored by clusters predicted by RareQ and a magnified view showing the hippocampus labeled by identified clusters. **b** Spatial expression of CA2 marker genes *Stard5* and *Plcxd3* in (**a**). **c** Spatial expression of CA1, DG and CA3 marker genes in (**a**). **d** Violin plot comparing the distribution of marker genes between CA3_1 and CA3_2 subsets. **e** UMAP projects of cells from the mouse hippocampus colored by RareQ clusters, with CA2 and CA3 subtypes highlighted. **f** Expression level of marker genes specific to CA2 and CA3 compartments. **g** Violin plots showing the expression level of marker genes identified in (**d**) across the CA2 and CA3 subtypes. **h** Heatmap displaying the expression level of the top 5 DEGs between CA2 and CA3 subtypes. **i** Violin plots visualizing the pathway enrichment score of selected pathways across the CA2 and CA3 subtypes. $p$-values were calculated using two-sided Wilcoxon test. Source data are provided as a Source Data file.

Compared to RareQ, existing algorithms—including scCAD, CellSIUS, FiRE, and GapClust—exhibited significant limitations in detecting rare spatial clusters. CellSIUS and GapClust failed to identify any rare spatial clusters, while FiRE incorrectly detected rare populations with

scattered, anatomically inconsistent distributions (Supplementary Fig. 61). Although scCAD successfully identified the CA2 and CA3 rare cell types, it missed the CA1 population and the subcompartments within CA3. Collectively, these results highlight RareQ as a powerful

and versatile tool for uncovering anatomical and molecular heterogeneity often overlooked by existing methods.

### RareQ identifies rare cells in spatial transcriptomics with paired protein data from renal cell carcinoma

To assess the efficacy of RareQ in analyzing multi-modal spatial data, we applied the algorithm to an in situ gene and protein expression dataset derived from formalin-fixed paraffin-embedded (FFPE) human RCC tissue available on the 10X Genomics website. This dataset encompassed 465,534 cells, profiled for the expression of 405 genes and 27 proteins. Following integration using WNN analysis, RareQ delineated 59 distinct cell clusters, including 42 rare populations (Supplementary Fig. 62a). These clusters were distinctly resolved in UMAP embeddings (Supplementary Fig. 62b). Annotation based on canonical markers identified clusters corresponding to tumor cells, T cells, myeloid cells, B cells (including plasma cells), fibroblasts, endothelial cells, and lymphatic endothelial cells (Supplementary Fig. 62c).

RareQ clustering revealed the presence of multiple TLS, comprising B cells, CD8$^+$ T cells, CD4$^+$ T cells, and endothelial cells, localized at the periphery of tumor regions (Supplementary Fig. 63a). Additionally, RareQ identified several rare myeloid subpopulations, including mast cells (clusters 2 and 4), cDCs (clusters 3 and 18), monocytes (cluster 6), and pDCs (cluster 10) (Supplementary Fig. 63b). Notably, the cDC population in cluster 18 exhibited elevated PD-L1 protein expression, consistent with our defined cDC subset in B-cell lymphoma. Furthermore, RareQ detected several rare proliferative populations (Supplementary Fig. 63c), with marker-based annotation identifying clusters 1 and 14 as proliferating tumor cells, cluster 5 as proliferating fibroblasts, cluster 7 as proliferating CD4$^+$ T cells, and cluster 12 as proliferating plasma cells (Supplementary Fig. 63d).

In the comparative analysis, scCAD and GapClust failed to identify these rare cell populations, while FiRE detected only proliferative tumor cells. And CellSIUS identified certain cell populations that were poorly resolved in UMAP embeddings (Supplementary Fig. 64). These findings highlight RareQ's capability to identify rare subpopulations in spatial transcriptomics with paired protein datasets.

## Discussion

Rare cell populations, although small in number, play disproportionately critical roles in tissue homeostasis, development, and disease progression. Their inherent scarcity and diverse spatial configurations pose significant challenges for detection using conventional clustering methods. A defining characteristic of these rare populations is their tendency to form tight-knit local neighborhoods in a low-dimensional kNN graph, often separated from major populations by either topological gaps or weak connectivity. RareQ distinguishes itself from local density estimation methods by characterizing the topological structure of the cell neighborhood graph. This enables identification of tightly connected cell cliques. By using high-$Q$ rare-cell cliques as stable anchors, RareQ directs label diffusion according to $Q$ values, making it particularly suited for identifying small, topologically self-organized subpopulations. This mechanism differs fundamentally from conventional methods that rely on uniform diffusion or modularity optimization (e.g., Louvain, Leiden). Simulation and real-world datasets consistently revealed that rare cells tend to have higher $Q$ values than major population cells, providing a strong discriminative signal even in the presence of high noise or subtle transcriptomic distinctions.

This graph-based approach distinguishes RareQ from existing methods such as CellSIUS, which relies on bimodal gene expression patterns within pre-defined clusters to define subpopulations[17], or GiniClust2, which ensembles clustering results weighted by feature importance[16]. Importantly, RareQ does not rely on prior identification of major cell types, rare cell-specific gene expression distributions, or external annotations, allowing for truly unsupervised discovery. This allows RareQ to overcome a key drawback of two-step clustering

methods, which often fail to detect rare cell types that are overshadowed within heterogeneous clusters. RareQ's strength lies not only in its sensitivity and specificity but also in its modality-agnostic design. In contrast to deep learning–based multi-omics integration methods like MarsGT—which rely on paired scRNA-seq and scATAC-seq data and may lose modality-specific signals due to forced alignment across heterogeneous omic layers, RareQ offers greater flexibility by analyzing each modality independently or integrating them using a WNN-based strategy, thereby preserving biologically meaningful modality-specific signals and avoiding signal dilution during integration. RareQ demonstrates robust performance in WNN-integrated multi-omics embeddings, effectively capturing rare populations that were otherwise missed by single-modality analysis or joint modeling approaches like MarsGT. From a computational perspective, RareQ exhibits superior scalability, processing datasets with over one million cells using minimal computational resources—an advantage over algorithms such as scCAD, EDGE, and GiniClust2, which often struggle with memory bottlenecks or prolonged runtimes when handling ultra-large datasets. Benchmarks against methods like FiRE and GapClust further confirmed that RareQ achieves a more favorable balance between speed, memory consumption, and accuracy in rare cell detection. RareQ also provides unique utilities in spatial transcriptomics analysis, an area where most existing rare cell detection tools underperform. In spatial datasets, RareQ not only recovered anatomically distinct rare populations, such as CA1, CA2, and CA3 hippocampal neurons, but also revealed fine-grained substructures—the spatial gradients of transcriptional activity within the CA3 region—that were not detected by tools like scCAD or FiRE. These findings demonstrate RareQ's potential to decode spatial architecture and identify region-specific rare cell states critical for tissue function.

Acknowledging limitations, the current RareQ framework does not explicitly integrate biological priors or gene-level functional annotations. Future developments may include hybrid models that synergize topological signals with supervised learning or gene signature-guided refinement. Additionally, for multi-modal data analysis, RareQ must be applied in parallel to each data modality and WNN-integrated embeddings to mitigate the risk of rare cell signals being obscured or diluted by averaging effects. This underscores the importance of future endeavors towards topology-aware, modality-sensitive integration strategies, potentially coupling RareQ with rare cell-aware embedding techniques for enhanced performance across diverse omics layers. Besides, in its current form, RareQ does not incorporate spatial proximity to improve the detection of spatially restricted rare cell populations, but this functionality could be introduced in future versions. In conclusion, RareQ offers an interpretable and computationally efficient solution for rare cell identification in single-cell and spatial transcriptomics. Its demonstrated scalability, cross-modality flexibility, and consistent ability to uncover biologically meaningful rare cell populations solidify its role as a valuable addition to the toolkit for dissecting cellular heterogeneity in complex tissues.

## Methods

### Data preprocessing

For scRNA-seq data, the raw count matrix was log2-normalized using Seurat R package[41] (version 4.3.0). Next, the top 2000 highly variable features were selected and scaled for principal component analysis (PCA). The $k$ nearest neighbors (k.param, default 20) for each cell were determined using FindNeighbors function (return.neighbor = TRUE) based on Euclidean distances computed in the embedded space of the top 50 principal components (PCs). Low-dimensional UMAP embeddings were also generated for visualization based on the top 50 PCs.

For scATAC-seq data, the raw peak matrix was normalized using TF-IDF in Signac R package[42] (version 1.9.0), and SVD was applied to derive a low-dimensional embedding. Like scRNA-seq data, the k.param (default 20) nearest neighbors for each cell were determined based on

Euclidean distances computed in the SVD embedded space. Notably, the first SVD component was excluded from downstream analysis due to high correlation with the number of fragments per cell[42].

For spatial data from EEL FISH and Xenium platforms, Seurat R package was used for normalization, scaling, and PCA-based dimension reduction. Due to the low number of features (fewer than 500 genes) in these data, we adjusted the scale factor to 80 and used all features as highly variable features in downstream analysis. The k.param (default 20) nearest neighbors for each cell were determined based on Euclidean distances in the low-dimensional PCA space.

For paired scRNA/ATAC-seq datasets, CITE-seq data, and spatial transcriptomics with paired protein abundance data, standard analytical workflows were applied to each modality independently. WNN analysis was then conducted to infer a WNN graph, denoting the most similar cells in the dataset based on a weighted combination of protein/ATAC and RNA similarities. Supervised principal component analysis (sPCA) was performed based on the WNN graph, identifying the set of principal components that could transform the data in a single modality to best capture the structure in a multimodal dataset. The k.param (default 20) nearest neighbors for each cell were determined based on Euclidean distances in the low-dimensional sPCA space. The whole process was accomplished using Seurat package.

For high-throughput single-cell data modalities beyond the workflow of Seurat, we prepared a helper function to construct a pseudo-Seurat object based on the low-dimensional embeddings derived by specialized tools. The k.param (default 20) nearest neighbors for each cell were determined based on Euclidean distances in this low-dimensional embedding space.

RareQ then took the processed Seurat object above as input to infer rare cell types, and outputs the predicted clusters. Thus, RareQ can be seamlessly incorporated into existing workflows by Seurat with a slight parameter change, making it convenient for downstream analysis. Moreover, RareQ accommodates emerging data modalities by constructing a pseudo-Seurat object from processed data.

## RareQ algorithm

RareQ highlights rare cell signals via curated $Q$ values, and further infers both major and rare cell populations by propagating the labels of waypoint cells with high $Q$ values to neighboring cells across the nearest neighbor graph. Specifically, RareQ can be summarized into the following procedures after data preprocessing.

1. Amplifying rare cell signals via $Q$ metrics. Specifically, RareQ takes the processed Seurat object as input and calculates the $Q$ values for each cell with a neighborhood size $k$ of 6, where a small value accommodates extremely rare cells. Let $\mathbf{S} = \{1, 2, \cdots, k\}$ be the set of $k$ cells including a cell and its $k$-1 nearest neighbors, its $Q$ value can be calculated as:

$$Q = \frac{\sum_{i,j \in \mathbf{S}} I_{i,j}}{k \times k} \tag{1}$$

$$I_{i,j} = \begin{cases} 1, & \text{An edge exists from } i \text{ to } j \\ 0, & \text{No edge exists from } i \text{ to } j \end{cases} \tag{2}$$

$I_{i,j}$ serves as an indicator, signifying the presence or absence of a connection from cell $i$ to $j$ in the kNN graph. If $i=j$, $I_{i,j}=1$. The denominator $k \times k$ measures the total number of directed edges from the $k$ cells to their $k$-1 nearest neighbors and themselves. $Q$ ranges between $1/k$ and 1, a high value implies dense internal connections within neighborhood.

2. Propagating cluster labels between neighboring cells by $Q$ values. Initially, each cell is treated as its own cluster. For each cell, it iteratively adopts the cluster label of its neighboring cell in $\mathbf{S}$ with the largest $Q$ value if its own $Q$ is lower. When a cell has multiple neighbors with the same highest $Q$ value, RareQ breaks the tie by

selecting the neighbor with the smallest cell index. Once convergence is reached or the maximum number of iterations is completed, the cluster labels of waypoint cells with local maxima of $Q$ values are propagated through the nearest neighbor graph, resulting in an initial set of clusters. Then the cluster label of each cell is further refined by iteratively applying a majority-voted prediction among itself and its $k$-1 nearest neighbors.

3. Recursively merging cell clusters. Cell clusters with average $Q$ values larger than 0.6 are retained as rare cell populations, while the remaining clusters are subject to a merging procedure to infer main cell types. Specifically, due to local variations of $Q$ values, propagating labels of suboptimal waypoint cells might introduce low-quality clusters, especially for main cell types. To merge these clusters, RareQ extends the $Q$ metric to evaluate the internal connectivity within clusters, denoted as $Q_c$. For a given cluster $C$, its neighboring cluster $C_n$ with the most connections from $C$ is first identified. If the $Q_c$ value of the combined cluster is greater than that of cluster $C$ and its neighboring cluster $C_n$, and the ratio of the number of connections from $C$ to $C_n$ to the total number of connections from cells in cluster $C$ exceeds 0.2, merging is performed. Let $\mathbf{C} = \{1, 2, \cdots, n\}$ be a cluster with $n$ cells, its $Q_c$ value can be calculated as:

$$Q_c = \frac{\sum_{i,j \in \mathbf{c}} I_{i,j}}{n \times k.param} \tag{3}$$

$$I_{i,j} = \begin{cases} 1, & \text{An edge exists from } i \text{ to } j \\ 0, & \text{No edge exists from } i \text{ to } j \end{cases} \tag{4}$$

$$k.param = \begin{cases} n, & n < 20 \\ 20, & n \geq 20 \end{cases} \tag{5}$$

Similar to $Q$ metric, $I_{i,j}$ serves as an indicator, signifying the presence or absence of a connection from cell $i$ to $j$ in the kNN graph. If $i=j$, $I_{i,j}=1$. The denominator $n \times k.param$ measures the total number of directed edges from the $n$ cells to their $k.param - 1$ nearest neighbors and themselves. $Q_c$ ranges between $1/k.param$ and 1, a high value implies dense internal connections within a cluster.

## Simulation of single-cell multi-omics data for benchmarking

Using paired human PBMC scRNA-seq and scATAC-seq data, we generated 150 simulated datasets following a similar strategy adapted from Wang et al.[22], with 50 replicates for each of the three simulation settings (Supplementary Table 1).

Sim-PBMC 1 represented a baseline simulation setting. One cell type was randomly selected from the two least abundant types in the original dataset to serve as the rare population, while the four most abundant cell types were retained as major populations. Approximately 5000 cells were sampled from these five cell types; the rare population was fixed at 50 cells, and the remaining cells were allocated among the four major populations according to their original proportions.

Sim-PBMC 2 increased the number of major populations to evaluate the method's performance in a more complex background. Here, the nine most abundant cell types were retained as major populations, and one was randomly selected from the two least abundant types as the rare population. As with Sim-PBMC 1, the rare population was fixed at 50 cells, with the remaining cells sampled from the nine major populations based on their original proportions, totaling approximately 5000 cells.

Sim-PBMC 3 increased the number of both rare and major populations to assess the method's sensitivity in detecting multiple rare cell types. In this setting, the ten most abundant cell types were retained as major populations, and five additional cell types with frequencies below 1% in the original dataset were randomly selected as rare populations. Approximately 5,000 cells were sampled from these 15 cell types according to their original proportions.

## In silico perturbation of scRNA-seq data for sensitivity evaluation

To analyze the sensitivity of RareQ to rare cell type identity, artificial scRNA-seq data was generated using the splatter R package[43]. The following command was used to generate this data: splatSimulate(group.prob = c(0.99, 0.01), method = 'groups', verbose = F, batchCells = 1500, de.prob = c(0.4, 0.4), out.prob = 0, de.facLoc = 0.4, de.facScale = 0.8, nGenes = 5000, seed = 2024).

The dataset consists of 1500 cells, each containing 5000 genes. Of these cells, 1487 represent the major cell type, while 13 define the minor type. After data preprocessing, Wilcoxon's rank sum test was used to identify DEGs with an FDR cutoff of 0.05 and an inter-group absolute fold-change cutoff of 1.5. The final 211 DEGs were preserved as a separate set. Meanwhile, another scRNA-seq dataset with the same number of cells (1500) and genes (5000) was simulated with the above command but using de.prob = c(0, 0), which served as the non-differential gene set.

The subsampled Jurkat dataset consists of 1556 cells, each containing 29,843 genes. Of these cells, 1540 represent the major 293 T cell type, while 16 define the minor Jurkat cell type. After data preprocessing, Wilcoxon's rank sum test was used to identify DEGs with an FDR cutoff of 0.05. To increase the number of DEGs for analysis, the inter-group absolute fold-change cutoff was adjusted to 1. The final 148 DEGs were preserved as a separate set. Additionally, 29,695 genes with a *p*-value exceeding 0.05 were retained as a distinct set of non-differential genes. For ablation analysis, we also subsampled 293 T cells from the major cell type and perturbed the expression of 10–100 genes by three-fold upregulation using the Jurkat dataset.

## Simulation of multi-scale scRNA-seq data for benchmarking

To analyze the scalability of RareQ to large data, a series of artificial scRNA-seq datasets were generated using the splatter R package. Similar to the above command, the batchCells and group.prob parameters were adjusted to generate datasets at different scales (1000, 2000, 5000, 10,000, 20,000, 50,000, 100,000, and 200,000 cells) with only 10 rare cells in each dataset.

## Statistics and reproducibility

In box plots, the horizontal line denotes the median value, the bottom and top of the box denote the lower and upper quartiles that represent the 25th (Q1) and 75th percentiles (Q3) of values, respectively. The interquartile range (IQR) is defined as the range between Q1 and Q3; whisker values are calculated as $Q1 - 1.5 \times IQR$ and $Q3 + 1.5 \times IQR$. DEGs were identified using the FindMarkers function in the Seurat R package (version 4.3.0), which employed a two-sided Wilcoxon rank sum test, with a false discovery rate (FDR) cutoff of 0.05. Fold-change values were calculated based on the mean expression levels of each gene between groups. *p*-values were adjusted using Bonferroni correction, accounting for the total number of genes in the dataset. Differential abundance test between AD and healthy controls in microglia clusters was applied using a negative binomial generalized linear model (NB-GLM) with quasi-likelihood (QL) estimation, consistent with the approach implemented in the Milo[44] algorithm.

No statistical method was used to pre-determine sample size. No data were excluded from the analysis; all genes in datasets were used throughout all analyses. The selection of pre-identified DEGs was randomized, all other experiments were not randomized. The investigators were blinded to allocation during experiments and outcome assessment.

## Comparison with other methods

Comparative analyses between RareQ and existing methods were conducted using default parameters. The evaluated tools and their associated preprocessing pipelines are summarized as follows: (i) FiRE (v1.0.1, GitHub), applied using default configurations. (ii) GapClust (v0.1.0, GitHub), data normalization was carried out using the scran R package for most datasets. For the large-scale mouse brain dataset, normalization was performed using Seurat to improve computational efficiency. (iii) CellSIUS (v1.0.0, GitHub), data were normalized using scran, with an initial clustering step implemented via the quickCluster function. (iv) RaceID (v0.3.5, CRAN), applied using default configurations. (v) GiniClust2 (v2.0, GitHub), applied with default settings. (vi) EDGE (v1.0, GitHub), data preprocessing and clustering based on EDGE embeddings were conducted using the Seurat package. (vii) scCAD (v1.0.0, GitHub), used with default settings as recommended by the authors. (viii) MarsGT (v1.0.0, GitHub), for benchmarking on paired scRNA-seq and scATAC-seq data from human PBMCs, the regulatory score matrix generated by MAESTRO[45] was obtained from Zenodo[46].

## Evaluation index

To evaluate the performance of existing rare cell identification tools, precision, recall, and $F_1$ score metrics were employed to quantify the accuracy in capturing each rare cell type. When evaluating the accuracy of clustering-based algorithms like RaceID, GiniClust2 and CellSIUS that predict both major and rare cell types simultaneously, we employed Normalized Mutual Information (NMI) to quantify the purity of the clustering output. Moreover, Jaccard index was also applied to measure the similarity between annotated rare cell types and predicted clusters. These metrics range from 0 to 1, where 1 provides the best performance. Before evaluation, each predicted cluster was annotated by the majority type of its constituting cells. For each rare cell type, the clusters with the same cell type annotation were considered as positive predictions.

Precision is defined as the proportion of correctly predicted rare cells relative to all rare cell predictions.

$$\text{Precision} = \frac{TP}{TP + FP} \quad (6)$$

Recall evaluates the proportion of correctly predicted rare cells from actual rare cells.

$$\text{Recall} = \frac{TP}{TP + FN} \quad (7)$$

$F_1$ score is a weighted average of precision and recall:

$$F_1 \quad \text{score} = 2 \times \frac{\text{Precision*Recall}}{\text{Precision} + \text{Recall}} \quad (8)$$

TP (True Positive) denotes the number of cells that are correctly predicted as rare cells and are actually rare cells. FP (False Positive) refers to the number of cells that are predicted as rare cells but are actually common cells. FN (False Negative) represents the number of cells that are predicted as common cells but are actually rare cells.

NMI measures the normalized dependency of the true labels on the predicted cluster, which was calculated using the aricode R package (version 1.0.2).

Jaccard index quantifies the membership similarity between predicted rare cells and true rare cells for each rare cell type. Let $\mathbf{S} = \{s_1, s_2, \cdots, s_S\}$ be the set of predicted clusters, and $\mathbf{T} = \{t_1, t_2, \cdots, t_T\}$ be the set of true labels.

$$\text{Jaccard index} = \frac{|\mathbf{S} \cap \mathbf{T}|}{|\mathbf{S} \cup \mathbf{T}|} \quad (9)$$

## Reporting summary

Further information on research design is available in the Nature Portfolio Reporting Summary linked to this article.

## Data availability

The 20 RNA-seq datasets for benchmarking in this study were obtained from various public websites under accession codes provided in Supplementary Table 2[47–63], including NCBI Gene Expression Omnibus (GEO) [https://www.ncbi.nlm.nih.gov/geo/], Cell Blast [https://cblast.gao-lab.org], Gut Cell Survey [https://www.gutcellatlas.org/], and 10X genomics [https://www.10xgenomics.com/datasets]. The 10 paired scRNA-seq and scATAC-seq datasets for benchmarking in this study were obtained from various public websites under accession codes provided in Supplementary Table 5, including scMMO-atlas[64] [https://www.biosino.org/scMMO-atlas/], scglue[65] example datasets [https://scglue.readthedocs.io/en/latest/data.html] and GEO. Human PBMC paired scRNA-seq and scATAC-seq data, 1.4 M mouse brain data, Alzheimer's disease data and mouse hippocampal data were obtained from GEO with accession codes GSE194122, GSE212606, GSE303823 and GSE261148. CITE-seq data of human bone marrow mononuclear cells was downloaded with SeuratData package (dataset identifier bmcite). The paired B lymphoma scRNA-seq and scATAC-seq dataset, Xenium spatial datasets of mouse brain and human RCC were obtained from the 10X Genomics website. Mouse brain spatial transcriptomic data on the EEL FISH platform was obtained from the Mouse Brain Atlas [http://mousebrain.org/adult/downloads.html]. Bulk AD RNA-seq data was downloaded from GEO with accession code GSE109887. Source data are provided with this paper.

## Code availability

RareQ is a user-friendly, efficient package developed in R. The source code for RareQ is available at GitHub [https://github.com/xiaolab-xjtu/RareQ] under MIT license. The source code is also archived in Zenodo[66] and accessible via [https://doi.org/10.5281/zenodo.17190972] [https://zenodo.org/records/17190972].

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

## Acknowledgements

We are grateful to Dr. Xuerui Yang for constructive discussions and to Dr. Penghua Lin for valuable assistance in data collection during the revision. This study was supported by National Key Research and Development Program of China (2021YFA1100702 to Z.X.), National Natural Science Foundation of China (32100535 and 32370706 to Z.X., 32200521 to B.F. and 82541006 to Y.X.), China Postdoctoral Science Foundation (2023M732808 to B.F.), the Chinese Academy of Medical Sciences (CAMS) Innovation Fund for Medical Sciences (CIFMS, 2023-I2M-C&T-B-021 to Dr. Penghu Lian), and Key Research and Development Plan of Shaanxi Province of China (2024SF-ZDCYL-02-04, W.Z.). Z.X. acknowledges support from the Youth Innovation Team Project of Xi'an Jiaotong University (xtr052025013, xtr052023008, and xtr052025012), and the Qinchuangyuan High-level Innovative Entrepreneurial Talent Project (QCYRCXM-2022-337).

## Author contributions

Z.X., B.F., and C.H. conceived and designed this project. B.F. implemented the RareQ. B.F., C.H., and Y.M. performed the analysis with critical support from W.Z. and Y.X. Z.X., B.F., and Y.M. wrote the paper. All authors have read and revised the final version of this paper.

## Competing interests

The authors declare no competing interests.
