## [Transparent Peer Review file · Nature Communications]

Cell Neighborhood Topology Directs Rare Cell Population Identification

Corresponding Author: Dr Zhengtao Xiao

Version 0:

Reviewer comments:

Reviewer #1

(Remarks to the Author)

Although the authors present the RareQ method for detecting rare cell populations in single-cell and spatial data, this work still falls short of the conceptual, technical, and biological standards expected by Nature Communications. Crucially, the study did not include disease-related datasets (e.g., cancer or neurodegeneration), making it impossible to assess the generalizability of the method. In addition, the benchmark design is nearly identical to MarsGT, which is of questionable novelty and fairness. Finally, for publicly available spatial transcriptomics datasets, RareQ's annotations used for comparisons do not match those in the original study reports, suggesting the possibility of misannotation or overclustering. Taken together, these shortcomings prevent me from recommending the manuscript for publication.

1. The manuscript initially specifies that rare cells are both functionally important and low in abundance, yet in the Results they are selected solely on abundance. This contradiction undermines the entire biological premise and renders subsequent claims unconvincing.
2. All data presented in the manuscript are derived exclusively from healthy donor samples, with no performance evaluation on datasets related to cancer or neurodegenerative diseases. This absence of validation on disease-relevant samples significantly limits the generalizability and practical applicability of the proposed method.
3. Ground-truth labels are taken at face value, and the study provides virtually no orthogonal evidence (e.g., immunostaining, lineage tracing, perturbation) to confirm the identity or significance of the purported rare populations. More regrettably, there is almost no in-depth functional investigation of the identified cell types, which substantially diminishes the scientific value and biological relevance of the study.
4. The benchmark experiments presented in Figure 2 largely overlap with those used in the previous study (MarsGT). However, MarsGT is included only as a baseline for the multi-omics integration task and is notably absent from the main benchmark and other experimental comparisons. Given that the benchmark settings in this manuscript closely align with those in MarsGT, excluding MarsGT from broader evaluations is unjustified and undermines the credibility and completeness of the performance assessment.
5. The separation of the two pDC subpopulations in Figure 4 relies on only subtle epigenomic differences, which are insufficient to justify defining them as distinct subtypes. Without clear evidence of functional divergence, the claim that these represent true subpopulations is unsubstantiated and premature.
6. The biological interpretation presented is inappropriate. In Figure 6, cluster C10 is enriched for the pathway "protein localization to the presynaptic region." Discussing the "presynaptic region" in the context of airway epithelial cells is highly unusual and seemingly irrelevant, which significantly undermines the reliability of the conclusions.
7. The analysis of the mouse brain (EEL FISH) dataset in Figure 7a–c presents several serious issues:
 - (1) RareQ's clustering results merely reproduce the specific structures highlighted in the original study, rather than uncovering novel biological insights. Furthermore, the annotation is less detailed—for instance, hippocampal subregions such as CA1/2 and CA3 are not distinguished, as they are in the original reference.
 - (2) Additionally, no baseline methods are included for comparison on this dataset, making it difficult to evaluate the distinct

advantages or effectiveness of RareQ in this context.

8. The discovery of rare cell types in Figure 7d appears to be driven largely by over-clustering. In the Xenium dataset, 162,000 cells were divided into 199 clusters—most containing fewer than 1% of cells. This suggests that the identification of rare populations stems more from arbitrary clustering granularity than from any unique strength of the method. Standard clustering tools could yield similar results, and the approach is highly sensitive to minor cellular heterogeneity, casting serious doubt on the biological validity of these rare cell types.

9. The description of the methodology on page 13, lines 468–477, is notably unclear and problematic.

(1) The parameter k mentioned here should be clearly distinguished from the k value defined in the Data Preprocessing section to avoid confusion.

(2) According to the manuscript's logic, since all cells in the set S are connected as nearest neighbors, this would imply that every cell is connected by edges, resulting in all calculated Q values being equal to 1.

10. Accurate identification of rare cells should be based on a correct overall clustering performance. However, the manuscript lacks evaluation metrics assessing the quality of the global clustering results.

11. Given the discussion on memory usage and runtime, it is necessary for the authors to specify the exact GPU and CPU models used in their experiments.

12. Since the study only uses spatial transcriptomics data for testing, the umbrella term “spatial omics” is misleading and should not be used. The authors should add at least one additional spatial modality, for example, spatial ATAC-seq, to substantiate multi-omic applicability.

13. In line 468, the parameter k is set directly to 6. Since k is a critical parameter, its impact on the experimental results should be explicitly discussed, with a dedicated analysis of how varying k affects performance.

14. The absence of example code in the provided code repository hinders reproducibility and makes it difficult for users to effectively apply the method.

(Remarks on code availability)

The absence of example code in the provided code repository hinders reproducibility and makes it difficult for users to effectively apply the method.

Reviewer #2

(Remarks to the Author)

RareQ is based on the idea of measuring fractional local connectivity (the so-called Q values in a k nearest neighbour sense) and associating rare groups to isolated clusters with high proportion of internal connections. This is a simple concept but practically sensible.

Clustering proceeds first by iterative assimilation through cluster label propagation and majority voting. This is then followed by a merging process.

The authors should comment on the sensitivity to key choices of parameters and how the values used in the analyses were chosen, specifically:

1. The average Q value threshold of 0.6 used to retain rare cell populations.
2. The ratio of connections from C to C_n to the total number of connections from cells in cluster C exceeding 0.2.
3. The $k = 20$ threshold.

The authors provide source code on Github and analysis data and scripts in Zenodo. However, I could not identify the scripts for the simulation analysis. It would be beneficial if the full simulation analysis was made available to give complete confidence in the results presented.

It would be beneficial for the authors to provide tables precisely describing the number of cell populations and numbers in their simulation scenarios. This is currently a little unclear from the descriptions in the manuscript.

Overall, the results appear promising but the reproducibility aspect of the work needs to be significantly stronger.

(Remarks on code availability)

The source code is provided but a full set of analysis scripts enabling the simulations to be reproduced could not be found.

Reviewer #3

(Remarks to the Author)

Identifying rare cell types from single-cell omics data has been a challenging task. This paper introduces a simplistic approach by propagating cluster labels on a knn graph according to the clustering coefficient, which seems to achieve significantly improved performance on the presented results. I find the description some of the presented evaluation metrics unclear, and it is hard to tell the general applicability and performance of the method due to a lack of 1) evaluations considering all cell types, and 2) a benchmark with single-cell data from diverse systems. Please see my detailed comments below.

Specific points:

1. The "Q value" is the same as the very widely used "clustering coefficient". Proper citation of the original paper (Watts, Duncan J., and Steven H. Strogatz. "Collective dynamics of 'small-world' networks." *nature* 393.6684 (1998): 440-442.) should be added. Throughout the article, it should not be described as a novel metric. Rather, it is more appropriate to put this as an application of a generally applicable classical index to the problem of single-cell data analysis.
2. It is described in Methods-RareQ algorithm-section 2, that the label propagation is stopped until convergence or a maximum number of iterations is reached. Since there are thousands of nodes in typical single-cell/spatial data but only 6 possible distinct Q values, I guess it is quite common for a node to have several highest Q value neighbors. I am therefore wondering that in this case, how do you break the tie? If a non-deterministic criteria is used, so that the convergence is then not guaranteed, is the result sensitive to the number of maximum iterations? If a deterministic criteria is used (e.g., always choose the cell with smaller index), is the result sensitive to criteria (in the example, will the result change much with perturbed indices)?
3. In the description of Evaluation index in the Methods section, the F1 score evaluates a binary classification task of whether a cell is a rare cell type or common cell type, according to lines 615 to 617. But the results (e.g., Fig.2) report cell type specific F1 scores. How are these computed exactly?
4. On line 605, it is described that "the clusters with the same cell type annotation were considered as positive predictions". In the evaluation, do you consider the situation where cells from the same rare cell type fall in different clusters? When there are multiple clusters whose major cell type are the same, are they all considered positive predictions? Since each cluster is annotated by the known cell type of majority of its constituting cells, in the extreme case, a clustering where each cell itself forms a cluster will all be considered positive predictions. And in a less extreme case, a high performance can be achieved under this metric by producing many small clusters.
5. Related to the previous point, it is important to report overall clustering performance as well, such as NMI over all cell types. In the current manuscript, NMI is reported on rare cell types only (Supp. Fig. 1).
6. Can the authors investigate more into whether the improved performance comes from the RareQ algorithm or due to better preprocessing of the data which led to higher quality distance metric among the cells? An added comparison to other methods using the the same preprocessing pipeline that RareQ used would be helpful to clarify this point.
7. For the spatial application, does the knn graph only rely on gene expression similarity? It would be interesting to explore the usage of both gene expression similarity and spatial proximity to reveal "spatially rare" cell types. That is, a spatially distinct region of a small number of cells from a type that is considered common from the single-cell perspective.
8. I appreciate the in-depth studies presented in the manuscript each focusing on one dataset (a total of three single-cell systems), for example, the hundreds of single-cell data simulated from the PBMC dataset. However, broad benchmark is also important to demonstrate the general applicability of RareQ. I suggest that the authors add a more comprehensive benchmark that includes real scRNA-seq datasets from diverse systems, using the numerous expert annotated single-cell data. Alternatively, at least including the datasets from the compared methods can be used to enrich the evaluation.

(Remarks on code availability)

Version 1:

Reviewer comments:

Reviewer #1

(Remarks to the Author)

Thank you for the revision. Despite the additional datasets and sensitivity analyses, the revision does not resolve several fundamental concerns raised in my original report, and the manuscript still falls short of the conceptual, methodological, and biological standards for Nature Communications.

1. The "Q value" corresponds to the classical clustering coefficient and reframes the work as an application of this index to rare cell detection. However, beyond adopting this metric, the manuscript still does not clearly articulate what is methodologically new relative to standard graph based clustering/label propagation pipelines (e.g., how RareQ's propagation/merging is fundamentally different from existing deterministic label propagation plus post hoc merging). Please provide the algorithmic innovations with ablations isolating the contribution of each step and benchmark these ablations directly.
2. The definition is "low abundance and biological significance evidenced by distinct molecular profiles". The supporting evidence largely relies on DEGs, GSEA, and signature scores, which are circular with respect to the same transcriptomic features used for discovery. No orthogonal validation (imaging, flow, perturbation, or independent pathological assessment) is provided.
3. The added lymphoma, AD snRNA seq, and RCC spatial multi omics datasets are welcome. However, the analyses do not demonstrate control for donor level effects, batch, or case/control imbalance, particularly for the AD microglia where cluster 15 is "enriched in AD". What is the statistical framework used to test group specific enrichment after accounting for donor as a random effect, and potential covariates; otherwise the disease association claims are difficult to interpret.

4. One of my important concerns was that RareQ's annotations on public spatial datasets did not match the original study reports. I do not see a systematic, side by side reconciliation with the original labels or a documented re annotation protocol.

5. CHOIR is applied asymmetrically, with no equivalent CHOIR validation for high resolution Seurat baselines under comparable features and parameters. Moreover, because CHOIR builds RF classifiers on the same expression space used to define clusters, there is a risk of self validation.

6. Were MarsGT hyperparameters tuned comparably? Were identical kNN graphs/latent spaces supplied where applicable? Were input features harmonized (e.g., WNN vs. integrated embeddings)?

7. In response to my request to avoid the umbrella term "spatial omics" if only RNA was used, you added one Xenium RNA+ADT dataset. This is helpful but still limited to a single sample and modality pair. Either temper the general claim (e.g., "spatial transcriptomics and one transcript protein dataset") or broaden the evidence (additional tissues/samples or orthogonal spatial modalities).

8. The tie break rule (smallest index) is deterministic but the biological stability of final clusters under index permutations is only assessed via aggregate metrics (F1/NMI).

9. In challenging spatial settings, RareQ may not add biological insight beyond recapitulating known structures.

10. To facilitate verification, please (i) include end to end scripts that reproduce a subset of the new disease analyses (e.g., AD microglia and RCC), (ii) pin package versions and random seeds, and (iii) provide exact commands for all figures cited in the response (R series).

After carefully reviewing the point-by-point response and new analyses, I find that the core concerns remain unresolved. In its current form, the manuscript is not suitable for publication in Nature Communications.

(Remarks on code availability)

Reviewer #2

(Remarks to the Author)

The authors have responded comprehensively to my comments.

The additional analyses would justify the choice of hyperparameters and should be included in the final manuscript.

(Remarks on code availability)

The Github repository has been updated to include the simulations. However, the quality of documentation around this and the tutorials is low and could be significantly improved to enable a competent person to reproduce the results in a reasonable amount of time. There are, for example, hard coded file paths and commented out lines in the simulation scripts. The effort that has been expended to enable a third party to read and understand the code is minimal.

Reviewer #3

(Remarks to the Author)

My comments have been properly addressed. I recommend the publication of the manuscript. (Zixuan Cang)

(Remarks on code availability)

Version 2:

Reviewer comments:

Reviewer #1

(Remarks to the Author)

This revised manuscript introduces RareQ, a graph-based clustering framework designed for rare-population discovery through Q-guided label propagation and Qc-guided recursive merging. While the authors have addressed some previous comments, several fundamental methodological concerns remain unresolved. Most notably, the justification for the method's performance and hyperparameter selection relies heavily on supervised metrics, which contradicts the "unsupervised discovery" premise. Furthermore, the distinction between RareQ and existing graph-clustering pipelines remains insufficiently demonstrated.

1. If RareQ is intended for unsupervised discovery, the reliance on label-dependent metrics (e.g., F1-score, NMI, Precision/Recall) to justify hyperparameter choices and "high-accuracy regions" is problematic. These metrics are unavailable in real-world discovery scenarios. Specifically, the suggestion in Supplementary Note 2 that users "modify parameters accordingly" to improve accuracy is impractical without ground-truth labels. The authors must provide a fully label-free parameter selection strategy and demonstrate its efficacy on benchmark datasets without invoking ground truth.

2. Given that only MarsGT appears to have undergone systematic tuning, while most baseline models run with default

settings, could the authors provide a predefined and fully fair benchmarking protocol? Furthermore, tuning should be conducted under the same search space/stopping criteria using unlabeled rules. Label-based metrics (e.g., F1/NMI) should not be used for tuning.

3. Regarding the EEL FISH dataset: The current explanation attributing discrepancies to "over-clustering" is insufficient. A concrete, step-by-step reconciliation analysis is required, including parameter sweeps and an identification of which preprocessing or resolution choices drive the divergence from original annotations.

4. The framework currently appears to be an operationalization of classical graph statistics (clustering coefficients) rather than a fundamentally new algorithmic contribution. The authors should explicitly formalize the mathematical or conceptual novelty that distinguishes RareQ from deterministic propagation. Please provide a direct comparison against strong, generic baselines (e.g., standard label propagation variants, or Leiden/Louvain with resolution control) using the exact same kNN graph. This is necessary to prove that RareQ's performance cannot be replicated by simple tuning of existing, widely-used pipelines.

5. Rare population discovery is highly sensitive to technical noise. How robust is RareQ to upstream choices (e.g., number of PCs, HVG selection, batch correction methods, or k)? A systematic analysis is needed to show if/when RareQ produces spurious clusters due to density gradients or batch effects. The authors should include "negative control" experiments (e.g., permuted data, batch-only simulations, or datasets without true rare types) to quantify the False Positive Rate (FPR) of the method. Practical diagnostics for users to identify potential failure modes in unlabeled settings should also be provided.

(Remarks on code availability)

Version 3:

Reviewer comments:

Reviewer #1

(Remarks to the Author)

The authors have provided additional analyses and clarifications in response to my previous review. Nevertheless, several key methodological issues remain only partially addressed.

1. The authors withdraw the earlier "over-clustering" statement and attribute the discrepancy to differing analysis goals. This, however, does not address my original request for a step-by-step reconciliation. Please clarify what is treated as the "ground truth"/annotation for EEL FISH in your evaluation, how the mapping/alignment between outputs and annotations is performed, and provide parameter sweeps that demonstrate what specifically drives agreement versus disagreement.

2. Same-graph comparisons using Louvain resolution sweeps are informative, but they are not sufficient to support the claim that the approach cannot be reproduced by standard graph pipelines. The authors should include same-graph comparisons with Leiden (resolution sweeps) and standard label propagation/LPA variants, with clearly specified stopping rules. In addition, key ablations (e.g., removing Q-guidance, removing Qc-based merging, or replacing Q with a closely related statistic) are needed to demonstrate which components are necessary for the reported gains.

(Remarks on code availability)

The code can be further optimized, and the description can also be more detailed.

REVIEWER COMMENTS

Reviewer #1 (Remarks to the Author):

Although the authors present the RareQ method for detecting rare cell populations in single-cell and spatial data, this work still falls short of the conceptual, technical, and biological standards expected by Nature Communications. Crucially, the study did not include disease-related datasets (e.g., cancer or neurodegeneration), making it impossible to assess the generalizability of the method. In addition, the benchmark design is nearly identical to MarsGT, which is of questionable novelty and fairness. Finally, for publicly available spatial transcriptomics datasets, RareQ's annotations used for comparisons do not match those in the original study reports, suggesting the possibility of misannotation or overclustering. Taken together, these shortcomings prevent me from recommending the manuscript for publication.

Response: We thank the reviewer for these critical comments. RareQ's conceptual basis is the Q value, a metric specifically designed to quantify the cliquishness (or interconnectedness) within a cell's k -Nearest Neighbors (k NN) network. Reviewer#3 highlights that this underlying concept is similar to the clustering coefficient used in a small-world network model (Watts & Strogatz, Nature, 1998; Kleinberg, Nature, 2000). Crucially, the novelty of our work is the adaptation of this classical network principle into a quantitative framework specifically for rare cell detection, not the benchmark design itself—since all methods are evaluated using the same benchmarking strategies and metrics for fairness.

In this revision, we have substantially expanded our evaluations to include a wide range of datasets, particularly disease-related datasets—including paired single-cell RNA and ATAC sequencing dataset from B cell lymphoma and single-nucleus RNA-seq dataset from Alzheimer's disease, as well as spatial omics datasets from renal cell carcinoma—to rigorously assess the generalizability of RareQ. We have carefully addressed all concerns, including the fairness of comparisons with MarsGT, annotation consistency, and potential over-clustering in public spatial datasets, as detailed in the point-by-point responses below.

We believe these comprehensive revisions address the reviewer's concerns and strengthen both the conceptual and biological contributions of this study. We respectfully ask the reviewer to reconsider and re-evaluate our revised manuscript.

1. The manuscript initially specifies that rare cells are both functionally important and low in abundance, yet in the Results they are selected solely on abundance. This contradiction undermines the entire biological premise and renders subsequent claims unconvincing.

Response: We appreciate this insightful comment, which helps us clarify the biological rationale and scope of our study on rare cells. We recognize that our

initial use of 'functionally important' may have created the impression that we experimentally validated the specific role of rare cells, which is not the primary focus of this computational study. To address this, we have refined our terminology: rare cells are now defined as subpopulations characterized by both low abundance and biological significance. The biological significance of these cells is evidenced by their distinct molecular profiles (e.g., unique transcriptomic, proteomic, or epigenomic signatures) that robustly differentiate them from dominant cell types. These unique profiles often suggest specialized roles, such as developmental intermediates, disease-driving subclones, or niche-specific populations. It is important to note that our method, RareQ, does not select cells based on a simple abundance threshold. Instead, it identifies cells based on their neighborhood relationships in low-dimensional space. This computational strategy ensures that the detected cells possess molecularly distinct gene expression profiles or occupy unique spatial niches, irrespective of their quantity.

To substantiate this biological relevance, we present extensive molecular evidence:

- ♦ **Molecular Signatures:** We conducted differential gene expression analyses and performed Gene Set Enrichment Analysis (GSEA) to link the rare cell signatures to biologically relevant processes and pathways.
- ♦ **Disease Relevance:** We included analyses of disease-related datasets (e.g., tumor and neurodegenerative contexts) that demonstrate the broad significance of the identified rare cells across diverse biological systems.
- ♦ **Independent Benchmarking:** We employed independent benchmarking methods, such as CHOIR, to verify that the detected rare cells are not statistical artifacts but exhibit molecular hallmarks consistent with specialized roles.

Taken together, these analyses demonstrate that RareQ detects rare cells with molecular features indicative of biological significance, thereby providing a crucial foundation for subsequent functional and mechanistic studies.

2. All data presented in the manuscript are derived exclusively from healthy donor samples, with no performance evaluation on datasets related to cancer or neurodegenerative diseases. This absence of validation on disease-relevant samples significantly limits the generalizability and practical applicability of the proposed method.

Response: Thank you for highlighting the need to evaluate RareQ on disease-relevant datasets to enhance its generalizability and practical applicability. In response, we have extended our analysis to include three diverse pathological datasets, demonstrating RareQ's robust performance in detecting biologically significant rare cell populations in cancer and neurodegenerative contexts. These results are now incorporated into the revised manuscript.

(1) **Lymphoma Dataset:** We applied RareQ to a matched single-cell RNA and ATAC sequencing dataset from a flash-frozen intra-abdominal lymph node tumor of a

patient with diffuse small lymphocytic lymphoma (also used in MarsGT's paper). RareQ identified rare cell populations with distinct marker genes—including plasmacytoid dendritic cells (pDCs), fibroblasts, endothelial cells, and plasma cells (Fig. R1)—substantially exceeding the performance of MarsGT (Fig. R2a). We focused on rare populations within the myeloid lineage: clusters 5 and 7 corresponded to cDC subtypes, clusters 12 and 15 to monocytes, and clusters 9 and 23 to macrophages (Fig. R2b, c). Notably, cluster 5 displayed elevated *PD-L1* expression and PD-1 checkpoint signaling activity, a feature previously linked to tumor immune suppression and checkpoint therapy response (Fig. R2d). Furthermore, both *PD-L1* and its upstream transcriptional regulators *STAT1* and *NFKB1* exhibited high expression and chromatin accessibility (Fig. R2e, f and Fig. R3), supporting their key regulatory role driving the transition of cDCs toward an immunotolerant phenotype.

Fig. R2 RareQ identifies rare myeloid cell types in B lymphoma. **a**, UMAPs visualizing cell clusters predicted by RareQ and MarsGT, with the common rare cell types identified by both highlighted. **b**, Dot plot demonstrates the expression value and proportion of the top differentially expressed genes between the myeloid cell clusters predicted by RareQ. **c**, Violin plot shows the expression of marker genes within the myeloid clusters for annotation. **d**, Gene signature scores of antigen presentation, macrophage activation, and PD-L1 pathways. P-values were calculated using ANOVA. Coverage plots for gene **(e)** *STAT1* and **(f)** *NFKB1*.

(2) **Alzheimer’s Disease Dataset:** Using single -nucleus RNA-sequencing data from cortical samples of non-cognitive impairment (NCI) controls and Alzheimer’s disease (AD) patients, RareQ identified multiple rare microglial and astrocytic subpopulations (Fig. R4a). Among microglia (marker genes *CSF1R*, *SPP1*), cluster 15 showed high expression of *IFI44L* and *NAMPT*, and was predominantly enriched in AD patients (Fig. R4b, c). Functional enrichment revealed strong activity of phagocytosis, myeloid leukocyte activation, and cytokine-mediated signaling pathways (Fig. R4d), suggesting an activated state with enhanced phagocytic capacity relevant to clearance of AD pathological products. Transcription factor analysis further showed enrichment of *STAT1*, *RUNX2*, and *ELF1* (Fig. R4e), consistent with an activated microglial phenotype. In addition, RareQ detected several rare astrocyte subpopulations (clusters 5, 12, 20) that were disproportionately enriched in AD and displayed highly reactive molecular features (Fig. R4f-i), indicating pathological activation. In contrast, alternative algorithms (scCAD, CellSIUS, RaceID, GiniClust2, FiRE, EDGE, and GapClust) failed to accurately identify these biologically significant populations (Fig. R5), underscoring RareQ’s superior sensitivity.

Fig. R4 RareQ identifies rare activated microglia and astrocyte populations in Alzheimer's Disease (AD). **a**, UMAP projections visualizing the major cell types and cell clusters predicted by RareQ. **b**, Dot plot illustrating the expression levels and proportions of top differentially expressed genes (DEGs) across microglial clusters predicted by RareQ. **c**, Bar plot showing the proportions of cells derived from AD patients and healthy control (CTRL) individuals within each microglial cluster. **d**, Bar plot displaying the enriched pathways of microglial clusters 15 and 16. **e**, Heatmap representing the inferred regulon activities across microglial clusters. **f**, Same as (b), but for astrocyte clusters. **g**, Same as (c), but for astrocyte clusters. **h**, Violin plot showing the expression of marker genes (reactive and homeostatic) within astrocyte clusters. **i**, Gene signature scores for the chaperone-mediated autophagy and regulation of amyloid fibril formation pathways. P-values were calculated using ANOVA.

(3) Renal Cell Carcinoma Dataset: In FFPE human renal cell carcinoma samples with in situ gene and protein (Antibody Derived Tag, ADT) expression data, RareQ discovered several spatially distinct rare cell types, including immune-cell aggregates forming tertiary lymphoid structures, myeloid subpopulations, and diverse proliferative subtypes (Fig. R6, 7). Comparative analyses again confirmed RareQ’s advantages in detecting rare, spatially organized cell types within disease-associated multi-omics data (Fig. R8).

These findings demonstrate RareQ’s ability to robustly identify rare cell populations with biological and clinical relevance across diverse pathological contexts. We have added these results to the Results and Discussion sections, strengthening the manuscript’s claims regarding RareQ’s applicability and impact.

Fig. R7 RareQ identifies rare myeloid and proliferative cell types in multiome spatial RCC data. **a**, Zoomed view of spatial maps of RCC data showing tertiary lymphoid structures (TLS) colored by RareQ clusters and surface protein markers. **b**, Dot plot illustrating the expression values and proportions of marker genes and proteins in rare clusters in the myeloid compartment. **c**, UMAP plot of rare cell clusters with high expression of Ki-67 protein highlighted. **d**, Dot plot illustrating the expression values and proportions of marker genes and proteins in highlighted rare clusters in (c).

3. *Ground-truth labels are taken at face value, and the study provides virtually no orthogonal evidence (e.g., immunostaining, lineage tracing, perturbation) to confirm the identity or significance of the purported rare populations. More regrettably, there is almost no in-depth functional investigation of the identified cell types, which substantially diminishes the scientific value and biological relevance of the study.*

Response: We thank the reviewer for emphasizing the importance of validating the identified rare cell populations. We agree that orthogonal experimental approaches—such as immunostaining, lineage tracing, or perturbation studies—would provide definitive confirmation of their identities and biological significance. However, as such experiments require specialized models, resources, and conditions beyond the scope of this computational study. We now provide a more comprehensive analysis of molecular signatures, enriched biological pathways, and corroborating evidence from published experimental studies to support the identities and relevance of the rare populations uniquely detected by RareQ. For example, for the two rare CA3 subpopulations identified in Xenium mouse brain data, we reference independent studies employing in situ hybridization, RNAscope, and riboprobe-based methods, all of which confirm the specificity of the marker genes

identified by RareQ (Fig. R9a-f). Moreover, we applied RareQ to analyze an independent scRNA-seq data (GSE261148) from mouse hippocampus, and successfully reidentified the CA3 subtypes with distinct expression profiles (Fig. R10a-e). Notably, markers (*Stard5*, *Strip2*, *Rspo2*, and *Nwd2*) discovered in the Xenium mouse brain data showed similar expression patterns within the CA3 subtypes in this scRNA-seq dataset (Fig. R10c).

These additional analyses strengthen the scientific foundation of our study and underscore RareQ's utility in uncovering rare populations that warrant future experimental investigation.

a Liu et al., Sci. Adv., 2024, Fig. 5A
Rspo2 in hippocampus

d Watanabe et al., Gene Expr. Patterns, 2023, Fig. 3S

b Liu et al., Sci. Adv., 2024, Fig. S7A

c Liu et al., Sci. Adv., 2024, Fig. S7C

e Yamada et al., Gene Expr. Patterns, 2022, Fig. 7G&H

f Verpoort et al., Dev. Cell, 2025, Figure S1E

Fig. R9 Representative image of marker genes *Rspo2*, *Nwd2* and *Plcx2* in published studies. a-d, Representative images of *Rspo2* *in situ* hybridization in mouse hippocampus. **e**, Mouse hippocampus stained with a *Nwd2* anti-sense riboprobe. **f**, Mouse coronal brain sections from the hippocampus probed for *Plcx2* (magenta) using RNAScope. The original publications were listed on top of plots.

4. The benchmark experiments presented in Figure 2 largely overlap with those used in the previous study (MarsGT). However, MarsGT is included only as a baseline for the multi-omics integration task and is notably absent from the main benchmark and other experimental comparisons. Given that the benchmark settings in this manuscript closely align with those in MarsGT, excluding MarsGT from broader evaluations is unjustified and undermines the credibility and completeness of the performance assessment.

Response: MarsGT was specifically designed for multi-omics datasets and, as noted in its original publication, cannot be applied to single-omics data. This inherent limitation precluded its inclusion in our initial single-omics benchmarks.

We have previously benchmarked RareQ against MarsGT in paired PBMC simulated and real datasets (Fig. R11a, b). To provide a more comprehensive evaluation, and ensure a fair and rigorous comparison, we have now incorporated 10 additional multi-omics datasets and performed benchmarking using multiple metrics. The results show that RareQ consistently outperforms MarsGT across various metrics, including F_1 score, precision, recall, and Normalized Mutual Information (NMI) (Fig. R11c, d). These expanded analyses underscore RareQ's superior robustness and effectiveness in multi-omics rare cell detection. We have presented these complete benchmarking results in the revised manuscript (Fig. 4).

5. The separation of the two pDC subpopulations in Figure 4 relies on only subtle epigenomic differences, which are insufficient to justify defining them as distinct subtypes. Without clear evidence of functional divergence, the claim that these represent true subpopulations is unsubstantiated and premature.

Response: We would like to clarify that the original Figure 4 only showed the top five loci with epigenomic differences. In the revised manuscript, we have now integrated transcriptomic, epigenomic, and regulatory analysis data to provide robust evidence for their distinct identities. Beyond the initially highlighted loci, our analysis identified 3618 differentially accessible regions ($p < 0.01$, absolute \log_2 fold

change ($|\log_2FC| > 1$) (Fig. R12a, b). The scRNA-seq analysis further revealed 48 differentially expressed genes ($p < 0.01$, $|\log_2FC| > 1$), including *IL3RA*, *KLF4*, *KLF6*, *BACH2*, *FYN*, and *NRP1*, that closely associated with pDC differentiation, function, or immune regulation (Fig. R12a).

In addition, SCENIC analysis identified distinct regulons, such as IKZF1, RFX5, REL, and FOXO3, which have clear associations with pDCs differentiation and function (Fig. R12c). Together, these detailed transcriptomic, epigenomic, and regulatory findings—now presented in Figure 4 and Supplementary Figure 35—provide strong evidence supporting the classification of these as genuinely distinct pDC subtypes.

6. The biological interpretation presented is inappropriate. In Figure 6, cluster C10 is enriched for the pathway “protein localization to the presynaptic region.” Discussing the “presynaptic region” in the context of airway epithelial cells is highly unusual and seemingly irrelevant, which significantly undermines the reliability of the conclusions.

Response: We thank the reviewer for highlighting this important point. We agree that this pathway is biologically irrelevant to airway epithelial cells and may have caused confusion. This artifact likely arose because some genes participate in multiple pathways, which can lead to the statistical enrichment of processes not directly related to the system under study.

To improve the biological interpretability of our results, we manually filtered the initial enrichment outputs to exclude pathways that, although statistically significant, are biologically implausible in the context of airway epithelial cells. This included removal of the “protein localization to the presynaptic region” pathway (Fig. R13).

7. The analysis of the mouse brain (EEL FISH) dataset in Figure 7a–c presents several serious issues:

(1) RareQ’s clustering results merely reproduce the specific structures highlighted in the original study, rather than uncovering novel biological insights. Furthermore, the annotation is less detailed—for instance, hippocampal subregions such as CA1/2 and CA3 are not distinguished, as they are in the original reference.

Response: We acknowledge that in this dataset RareQ did not generate new biological discoveries. The original study reported 188 highly fine-grained subpopulations, distinguishing CA1/2 from CA3 within the hippocampus, which RareQ did not separate. We speculate that the authors may have used additional information to aid in their classification, as we were unable to reproduce these distinctions even when using more sensitive parameters to re-cluster CA subtypes. We've repeatedly tried to reach the authors for clarification, but haven't received a response.

Upon closer examination, we found that the reported differences between the original CA3 and CA1/2 populations were minimal, with *Cpne4* being the only gene showing a statistically significant difference (Fig. R14a, b). Moreover, benchmarking the original clustering with CHOIR (Cluster Hierarchy Optimization by Iterative Random forests) (<https://doi.org/10.1038/s41588-025-02148-8>), a recently published statistical framework specifically designed to guard against both over- and under-clustering in single-cell data, indicated that many of these fine-grained clusters likely reflected over-clustering rather than biologically distinct populations (Fig. R14c).

Despite this, RareQ still outperformed existing rare cell detection methods on this dataset, demonstrating clear technical advantages (Fig. R15). Since no novel biological insights were derived here, we have moved these results to the supplementary file (now Supplementary Figure 52).

(2) Additionally, no baseline methods are included for comparison on this dataset, making it difficult to evaluate the distinct advantages or effectiveness of RareQ in this context.

Response: We analyzed the dataset using several baseline methods—scCAD, CellSIUS, FiRE, and GapClust—for comparison. Neither scCAD nor CellSIUS detected any rare cell clusters. Although FiRE and GapClust identified some rare clusters, their spatial distributions were highly dispersed and lacked meaningful anatomical structure (Fig. R15a, b). These findings suggest that existing methods struggle to reliably identify rare cell populations and assign them to their correct biological locations. In contrast, RareQ accurately identified and localized rare cells within the rostral migratory stream, the hippocampus, and the Purkinje layer of the cerebellum. This demonstrates RareQ’s distinct advantage in both detecting and

spatially resolving rare cell populations within their proper anatomical context. These comparative results have been incorporated into the revised manuscript.

8. *The discovery of rare cell types in Figure 7d appears to be driven largely by over-clustering. In the Xenium dataset, 162,000 cells were divided into 199 clusters—most containing fewer than 1% of cells. This suggests that the identification of rare populations stems more from arbitrary clustering granularity than from any unique strength of the method. Standard clustering tools could yield similar results, and the approach is highly sensitive to minor cellular heterogeneity, casting serious doubt on the biological validity of these rare cell types.*

Response: To ensure that RareQ's identification of rare populations was not an artifact of over-clustering, we applied CHOIR (Cluster Hierarchy Optimization by Iterative Random forests) (<https://doi.org/10.1038/s41588-025-02148-8>), a recently published statistical framework specifically designed to guard against both over- and under-clustering in single-cell data. Using CHOIR's compareClusters function, we evaluated the 199 clusters identified by RareQ in the Xenium dataset. CHOIR

confirmed that 189 of these clusters represented statistically distinct populations (Fig. R16a), indicating that the vast majority of RareQ-defined clusters are biologically meaningful rather than spurious subdivisions.

Importantly, RareQ is not simply a high-resolution clustering approach; it leverages local neighborhood connectivity (Q) to distinguish genuine rare populations from arbitrary splits. To further validate this, we compared RareQ with Seurat's Louvain clustering across a wide range of resolutions (0.8, 1, 2, 5, and 10). At default and moderate resolutions (0.8–2), Seurat failed to detect the CA2 region (Fig. R16b). At higher resolutions (5 and 10), Seurat recovered CA2 but was unable to resolve the CA3_1 and CA3_2 subpopulations, which RareQ successfully distinguished.

Together, these results—combining CHOIR's statistical validation with comparative benchmarking—demonstrate that RareQ can identify rare populations with high specificity while maintaining biological plausibility.

9. The description of the methodology on page 13, lines 468–477, is notably unclear and problematic.

(1) The parameter k mentioned here should be clearly distinguished from the k value defined in the Data Preprocessing section to avoid confusion.

Response: We appreciate the reviewer’s observation. In the Data Preprocessing section, the k value refers to the parameter used for constructing the k NN graph, whereas in RareQ, the k value denotes the parameter used in calculating the Q

value. To avoid confusion, we have renamed the former as k.param and added further clarification to clearly distinguish between these two definitions.

(2) According to the manuscript's logic, since all cells in the set S are connected as nearest neighbors, this would imply that every cell is connected by edges, resulting in all calculated Q values being equal to 1.

Response: We thank the reviewer for this comment. It's important to clarify that Q will equal 1 only when a cell and its k-nearest neighbor network form a perfect cliquishness, meaning every member is connected exclusively to every other member in the set S (Fig. R17a). However, when there exist edges from cells in S to external cells (i.e., outward connection (OC) > 0) (Fig. R17b, c), the Q value would be less than 1, as the denominator $|IC| + |OC|$ is always greater than the numerator $|IC|$.

10. Accurate identification of rare cells should be based on a correct overall clustering performance. However, the manuscript lacks evaluation metrics assessing the quality of the global clustering results.

Response: We thank the reviewer for raising this important point. In our initial submission, we primarily evaluated performance using metrics widely applied in published rare cell detection studies, such as F_1 score. While we did assess global clustering quality on the PBMC simulation scRNA-seq datasets (Fig. R18a), we acknowledge that this evaluation was not extended to all datasets.

To address this, we have now systematically evaluated the overall clustering performance using the Normalized Mutual Information (NMI) metric across a broader

range of datasets. On one hand, we have benchmarked RareQ against the clustering algorithms (CellSIUS, RaceID, GiniClust2, EDGE) tailored for scRNA-seq dataset in the four PBMC scRNA-seq datasets and 20 real scRNA-seq datasets (Fig. R18b, c). On the other hand, we compared RareQ with multi-omics approach MarsGT in the 150 paired PBMC simulation datasets, 4 PBMC multiome datasets, and 10 paired scRNA-seq and scATAC-seq datasets (Fig. R19a, b and Fig. R11d). As shown in the new results, RareQ consistently achieved the highest NMI scores compared with other algorithms, demonstrating both accurate rare cell identification and robust global clustering performance. These results have been incorporated into the revised manuscript (Supplementary Figure 22, 46).

11. Given the discussion on memory usage and runtime, it is necessary for the authors to specify the exact GPU and CPU models used in their experiments.

Response: Thank you for this helpful suggestion. We have added detailed hardware specifications to the revised manuscript. In most experiments, we used a computing platform with 64 CPUs (Intel® Xeon® Gold 6226R, 2.90 GHz), 750 GB of memory, and 2 GPUs (NVIDIA RTX A6000, 48 GB memory each). For benchmarking MarsGT, we employed a platform with 128 CPUs (AMD EPYC 7543, 32-Core Processor).

12. Since the study only uses spatial transcriptomics data for testing, the umbrella term “spatial omics” is misleading and should not be used. The authors should add at least one additional spatial modality, for example, spatial ATAC-seq, to substantiate multi-omic applicability.

Response: We thank the reviewer for raising this point. To address the concern, we extended our analysis to an *in situ* gene–protein expression dataset derived from a human renal cell carcinoma FFPE sample by the Xenium platform.

Using RareQ, we identified 59 distinct cell clusters forming complex spatial architectures (Fig. R6a). These clusters were clearly separated in UMAP space, reflecting unique transcriptional and proteomic signatures, including major populations such as B cells, T cells, myeloid cells, tumor cells, fibroblasts, and endothelial cells (Fig. R6b, c).

RareQ further uncovered several biologically meaningful features:

- **Tertiary lymphoid structures (TLS):** In the peritumoral region, RareQ resolved highly organized aggregates of B cells consistent with TLS (Fig. R7a).
- **Myeloid heterogeneity:** RareQ distinguished diverse myeloid subpopulations, including mast cells, conventional dendritic cells (cDCs), plasmacytoid dendritic cells (pDCs), mature cDCs, and monocytes, each defined by distinct gene- and protein-level markers (Fig. R7b).
- **Rare proliferative subclusters:** RareQ detected rare Ki-67⁺ proliferative clusters (clusters 1, 5, 7, 12, and 14) across tumor, fibroblast, CD4⁺ T-cell, and plasma cell lineages (Fig. R7c, d).

In contrast, methods such as CellSIUS, FIRE, and GapClust—when applied to gene expression data alone—failed to resolve subclusters with comparable resolution (Fig. R8).

Together, these results demonstrate RareQ's capacity to analyze spatial multi-omic datasets with high precision.

13. In line 468, the parameter k is set directly to 6. Since k is a critical parameter, its impact on the experimental results should be explicitly discussed, with a dedicated analysis of how varying k affects performance.

Response: We thank the reviewer for this valuable suggestion. To address this, we conducted a comprehensive sensitivity analysis by systematically varying k from 5 to 30 across 150 simulated and 20 real scRNA-seq datasets.

Our analysis showed that RareQ's performance is highly robust to changes in k . In the simulated PBMC datasets, both the F_1 scores (measuring rare cell detection accuracy) and NMI values (measuring overall clustering quality) remained stable with only minimal fluctuations (Fig. R20a, b). We observed the same trend in the 20 real datasets, where the F_1 , precision, recall, and NMI values consistently stayed high across the entire range of k values (Fig. R20c, d).

These results demonstrate that RareQ is not overly sensitive to the choice of k , and its performance remains stable across a wide range of values. We have included these findings in the revised manuscript.

14. The absence of example code in the provided code repository hinders reproducibility and makes it difficult for users to effectively apply the method.

Response: We have updated the code repository (www.github.com/xiaolab-xjtu/RareQ) to include example code, sample datasets, and a step-by-step tutorial.

Reviewer #1 (Remarks on code availability):

The absence of example code in the provided code repository hinders reproducibility and makes it difficult for users to effectively apply the method.

Response: Addressed in our response to Comment 14.

Reviewer #2 (Remarks to the Author):

RareQ is based on the idea of measuring fractional local connectivity (the so-called Q values in a k nearest neighbour sense) and associating rare groups to isolated clusters with high proportion of internal connections. This is a simple concept but practically sensible.

Clustering proceeds first by iterative assimilation through cluster label propagation and majority voting. This is then followed by a merging process.

The authors should comment on the sensitivity to key choices of parameters and how the values used in the analyses were chosen, specifically:

1. The average Q value threshold of 0.6 used to retain rare cell populations.

Response: We thank the reviewer for raising this concern. We selected an average Q-value threshold of 0.6 as the default because it balances the detection of rare cell populations with the overall accuracy of cell clustering. To evaluate its robustness, we performed a comprehensive sensitivity analysis by varying the threshold from 0.1 to 0.9 using both 150 simulated and 20 real single-cell RNAsequencing datasets.

In the three sets of PBMC simulation datasets, F_1 scores (reflecting the accuracy of rare cell detection) showed only a slight decline around a threshold of 0.6 (Fig. R21a). In contrast, NMI values (reflecting overall clustering quality) exhibited a modest increase at 0.4 (Fig. R21b). Across 20 real datasets, F_1 , precision, and recall remained stable across thresholds, while NMI again showed a small increase at 0.4 but stabilized around 0.6 (Fig. R21c&d).

These findings highlight a clear trade-off: lower Q thresholds slightly favor rare cell detection, whereas higher thresholds better preserve overall clustering accuracy. We therefore chose 0.6 as the default threshold, as it provides an effective balance between these two objectives and delivers robust performance across datasets. We have also noted in the revised manuscript that users may adjust this parameter to align with their specific research goals.

2. The ratio of connections from C to C_n to the total number of connections from cells in cluster C exceeding 0.2.

Response: This ratio serves as the threshold for merging clusters. To evaluate its effect, we applied RareQ with values ranging from 0.1 to 0.9 across both simulated and real scRNA-seq datasets.

In the PBMC simulation datasets, both F_1 scores and NMI remained high and showed minimal sensitivity to changes in this ratio (Fig. R22a, b). In the real scRNA-seq datasets, the average F_1 score, precision, and recall were also stable (Fig. R22c, d); however, we observed higher variance at a ratio of 0.1 (Fig. R22c), suggesting that

performance may be less reliable under this setting.

Taken together, these results indicate that setting the ratio exceeding 0.2 provides stable and reliable performance across most scenarios.

3. The $k = 20$ threshold.

Response: This parameter, now renamed as $k.param$, represents the number of nearest neighbors in data preprocessing. We applied RareQ using a $k.param$ range of 10 to 30 across both simulated and real scRNA-seq datasets.

In the PBMC simulation datasets, both F_1 scores and NMI remained high, showing only a very slight decline as $k.param$ increased (Fig. R23a, b). In the real scRNA-seq datasets, increasing $k.param$ had minimal effect on performance metrics (Fig. R23c, d). Based on these results, we recommend using a moderate $k.param$ value, such as 20, which provides optimal and stable performance.

Taken together, these evaluation results have been incorporated and discussed in the revised manuscript.

The authors provide source code on Github and analysis data and scripts in Zenodo. However, I could not identify the scripts for the simulation analysis. It would be beneficial if the full simulation analysis was made available to give complete confidence in the results presented.

Response: We have provided the scripts for the simulation analysis in the repository (www.github.com/xiaolab-xjtu/RareQ), along with example code, sample datasets, and a step-by-step tutorial (as suggested by Reviewer #1), to ensure full reproducibility and transparency.

It would be beneficial for the authors to provide tables precisely describing the number of cell populations and numbers in their simulation scenarios. This is currently a little unclear from the descriptions in the manuscript.

Response: We have listed the number of cell populations and corresponding cell counts in each simulation scenario in the supplementary table 1.

Overall, the results appear promising but the reproducibility aspect of the work needs to be significantly stronger.

Response: We appreciate the reviewer's recognition of the promise of our results and the emphasis on reproducibility.

Reviewer #2 (Remarks on code availability):

The source code is provided but a full set of analysis scripts enabling the simulations to be reproduced could not be found.

Response: This concern has been addressed as noted in our earlier response.

Reviewer #3 (Remarks to the Author):

Identifying rare cell types from single-cell omics data has been a challenging task. This paper introduces a simplistic approach by propagating cluster labels on a knn graph according to the clustering coefficient, which seems to achieve significantly improved performance on the presented results. I find the description some of the presented evaluation metrics unclear, and it is hard to tell the general applicability and performance of the method due to a lack of 1) evaluations considering all cell types, and 2) a benchmark with single-cell data from diverse systems. Please see my detailed comments below.

Response: We thank the reviewer for this thoughtful and constructive feedback. In the revised manuscript, we have provided clearer definitions and justifications for each evaluation metric, reported the method's performance across all cell types, and expanded our benchmarking to include additional datasets from diverse biological systems. We are grateful for the reviewer's suggestions, which have

helped us improve both the rigor and clarity of our work.

Specific points:

*1. The “Q value” is the same as the very widely used “clustering coefficient”. Proper citation of the original paper (Watts, Duncan J., and Steven H. Strogatz. "Collective dynamics of ‘small-world’ networks." *nature* 393.6684 (1998): 440-442.) should be added. Throughout the article, it should not be described as a novel metric. Rather, it is more appropriate to put this as an application of a generally applicable classical index to the problem of single-cell data analysis.*

Response: We thank the reviewer for this valuable comment. We have now revised the manuscript to present the Q value as an application of this classical network theory concept rather than a novel metric. We have also cited this work to provide the theoretical background underlying the design of the RareQ method.

2. It is described in Methods-RareQ algorithm-section 2, that the label propagation is stopped until convergence or a maximum number of iterations is reached. Since there are thousands of nodes in typical single-cell/spatial data but only 6 possible distinct Q values, I guess it is quite common for a node to have several highest Q value neighbors. I am therefore wondering that in this case, how do you break the tie? If a non-deterministic criteria is used, so that the convergence is then not guaranteed, is the result sensitive to the number of maximum iterations? If a deterministic criteria is used (e.g., always choose the cell with smaller index), is the result sensitive to criteria (in the example, will the result change much with perturbed indices)?

Response: In the RareQ algorithm, we use a deterministic approach to handle ties: when a node has multiple neighbors with the same highest Q value, we break the tie by selecting the neighbor with the smallest cell index.

To ensure our results are not sensitive to this choice, we conducted a robust analysis. We performed 30 iterations of shuffling the cell indices of both simulated and real scRNA-seq datasets. Our results showed that the performance of RareQ remained highly stable across all shuffled datasets (Fig. R24). There were only subtle variations in the evaluation metrics, confirming that the deterministic tie-breaking criterion does not significantly impact the final outcome.

3. In the description of Evaluation index in the Methods section, the F₁ score evaluates a binary classification task of whether a cell is a rare cell type or common cell type, according to lines 615 to 617. But the results (e.g., Fig.2) report cell type specific F₁ scores. How are these computed exactly?

Response: For datasets with a single rare cell type, the F₁ score is calculated using standard binary classification metrics.

For datasets containing multiple rare cell populations, we compute an F₁ score for each population individually. This is done by treating each rare cell population as

the positive set in turn, with all other cells considered the negative set. This approach allows for an independent performance assessment of each rare cell type, preventing populations with lower F_1 scores from being masked by those with higher scores.

To illustrate this point, consider a hypothetical dataset with two imbalanced rare cell types: Type A (50 cells) and Type B (10 cells). The table below shows how a single overall F_1 score can obscure poor performance on the smaller population, while our individual F_1 score calculation provides a more transparent evaluation. In Scenario 2 and 3, an overall F_1 score remains high, masking the poor detection of Type B. By contrast, our method's individual F_1 scores accurately reveal the specific performance for each cell type, providing a more balanced and accurate evaluation.

In the revised manuscript, we also added Normalized Mutual Information (NMI) to provide a more comprehensive evaluation of clustering performance.

Scenarios	Number of rare cells detected		Overall F_1 Score	Individual F_1 score	
	Type A (n=50)	Type B (n=50)		Type A	Type B
1	50/50	10/10	1.00	1.00	1.00
2	50/50	5/10	0.95	1.00	0.66
3	50/50	0/10	0.90	1.00	0.00

4. On line 605, it is described that “the clusters with the same cell type annotation were considered as positive predictions”. In the evaluation, do you consider the situation where cells from the same rare cell type fall in different clusters? When there are multiple clusters whose major cell type are the same, are they all considered positive predictions? Since each cluster is annotated by the known cell type of majority of its constituting cells, in the extreme case, a clustering where each cell itself forms a cluster will all be considered positive predictions. And in a less extreme case, a high performance can be achieved under this metric by producing many small clusters.

Response: Thank you for raising this important point regarding the risk of over-clustering skewing performance metrics. We agree that our initial metrics, such as the F_1 -score, did not sufficiently penalize over-clustering. To address this, we've now incorporated standard global clustering metrics, Normalized Mutual Information (NMI). Both of these metrics explicitly penalize over-clustering, as a perfect score (1.0) requires that cells from a single true type are grouped into a single predicted cluster. Additionally, for real datasets we validate clusters using marker gene expression to ensure that each cluster corresponds to a biologically distinct population, as over-clustered groups generally lack unique markers. These combined strategies confirm that RareQ avoids inflated performance and produces meaningful clusters.

5. Related to the previous point, it is important to report overall clustering performance as well, such as NMI over all cell types. In the current manuscript, NMI is reported on rare cell types only (Supp. Fig. 1).

Response: We thank the reviewer for this helpful suggestion. As also noted by Reviewer #1, we have extended our evaluation to include overall clustering performance on both the current benchmark datasets and 20 additional real scRNA-seq datasets, spanning single-transcriptomic and multi-omic settings (Fig. R18, 19, and Fig. R11d). In all comparisons, RareQ consistently achieved higher NMI scores than the other methods, confirming its strong overall clustering performance.

6. Can the authors investigate more into whether the improved performance comes from the RareQ algorithm or due to better preprocessing of the data which led to higher quality distance metric among the cells? An added comparison to other methods using the the same preprocessing pipeline that RareQ used would be helpful to clarify this point.

Response: Thank you for this valuable comment. It is crucial to distinguish between algorithmic performance and the effects of data preprocessing. We confirm that RareQ does not rely on any specialized preprocessing steps to enhance its performance. All datasets were processed using the standard Seurat pipeline, including normalization, scaling, PCA, and construction of the *k*NN graph in low-dimensional space, with no additional steps introduced.

For fairness, EDGE was benchmarked using the same Seurat preprocessing workflow as RareQ. Similarly, Python-based methods such as scCAD and MarsGT were run within the Scanpy framework but with equivalent preprocessing steps. The remaining methods (RaceID, FiRE, GapClust, GiniClust2, and CellSIUS) typically employ their own built-in preprocessing, but in this revision, we also re-ran them using the Seurat pipeline. Across both simulated and 20 real datasets, RareQ consistently outperformed all other methods, confirming that its superior performance arises from the algorithmic design rather than differences in preprocessing (Fig. R25).

7. For the spatial application, does the knn graph only rely on gene expression similarity? It would be interesting to explore the usage of both gene expression similarity and spatial proximity to reveal “spatially rare” cell types. That is, a spatially distinct region of a small number of cells from a type that is considered common from the single-cell perspective.

Response: We agree with your point and appreciate the suggestion. In our initial analysis of spatial omics data, the k NN graph was constructed solely on gene expression similarity. To address the reviewer’s recommendation, we extended our analysis by applying a Weighted Nearest Neighbor (WNN) approach to integrate k NN graphs constructed from both gene expression and spatial proximity (Fig. R26a). Our results showed that this method was effective in identifying spatially rare cell types (i.e., cells belonging to the same cluster from a single-cell perspective but located in spatially distinct regions; (Fig. R26b). However, it sometimes obscured subtle transcriptional differences between spatially adjacent rare populations, such

as CA3 subtypes (Fig. R27). This finding highlights a critical trade-off: a strong emphasis on spatial proximity can inadvertently merge transcriptionally distinct but spatially close cell types, thereby masking important biological distinctions. Acknowledging that the optimal balance between these two data modalities may vary across datasets, we plan to further evaluate the strategy for adjusting the weighting of spatial proximity within the WNN framework. We will incorporate this refined approach as an optional parameter in the next release of RareQ, allowing users to fine-tune the method based on their specific biological questions.

8. I appreciate the in-depth studies presented in the manuscript each focusing on one dataset (a total of three single-cell systems), for example, the hundreds of single-cell data simulated from the PBMC dataset. However, broad benchmark is also important to demonstrate the general applicability of RareQ. I suggest that the authors add a more comprehensive benchmark that includes real scRNA-seq datasets from diverse systems, using the numerous expert annotated single-cell data. Alternatively, at least including the datasets from the compared methods can be used to enrich the evaluation.

Response: We thank the reviewer for this valuable suggestion. In the revised manuscript, we have expanded our benchmarking analyses to include 20 additional expert-annotated real scRNA-seq datasets spanning diverse biological systems, such as worm neurons, mouse airway, brain, heart, retina, mammary gland, kidney, and human lymph node, kidney tumor, liver, pancreas, and gut (Fig. 2d, e). These

datasets were generated using multiple platforms, including 10x Genomics, Smart-seq2, Drop-seq, and inDrop.

To demonstrate its broad applicability on multi-omics data, we further benchmarked RareQ against MarsGT using 10 paired scRNA-seq and scATAC-seq datasets from diverse systems (e.g., mouse brain, kidney, colon, immune cells, human PBMC, brain, and retina). RareQ consistently outperformed existing methods, achieving higher F1 scores, precision, and recall (Fig. 4e, f).

In addition, as noted by Reviewer #1, we have incorporated three disease-related datasets in this revision to demonstrate RareQ's robustness in clinically relevant contexts. Collectively, these expanded analyses provide strong evidence of RareQ's broad applicability and robustness across diverse systems, technologies, and conditions. A full list of datasets used in this study is now provided in Supplementary Table 2 and 5.

REVIEWER COMMENTS

Reviewer #1 (Remarks to the Author):

Thank you for the revision. Despite the additional datasets and sensitivity analyses, the revision does not resolve several fundamental concerns raised in my original report, and the manuscript still falls short of the conceptual, methodological, and biological standards for Nature Communications.

Response:

We thank the reviewer for the rigorous standards and constructive critique, which was essential for enhancing the quality of our manuscript. We have made extensive revisions to improve the methodological clarity of RareQ and to more effectively illustrate its capacity to yield novel scientific insights. These include ablation studies to elucidate the algorithmic innovations, integration of additional biological validation datasets, and implementation of reproducible analysis scripts to ensure transparency and replicability. We are confident that these changes now align with the rigorous criteria for publication in *Nature Communications*.

1. The “Q value” corresponds to the classical clustering coefficient and reframes the work as an application of this index to rare cell detection. However, beyond adopting this metric, the manuscript still does not clearly articulate what is methodologically new relative to standard graph based clustering/label propagation pipelines (e.g., how RareQ’s propagation/merging is fundamentally different from existing deterministic label propagation plus post hoc merging). Please provide the algorithmic innovations with ablations isolating the contribution of each step and benchmark these ablations directly.

Response:

We thank the reviewer for raising these concerns regarding the methodological novelty of RareQ and for providing the opportunity for further clarification. While the Q value is indeed inspired by the classical clustering coefficient in small-world network theory, RareQ transforms this concept into a dynamic, cell-specific topological measure that quantifies the interconnectivity within each cell’s neighborhood network. It is also extended to the cluster level (denoted Q_c) to guide post-hoc merging for optimizing clustering resolution and preventing over-clustering.

It is noteworthy that several established rare-cell identification algorithms—such as GiniClust, RaceID, and FiRE—also adapt well-established concepts to the single-cell context, such as Gini impurity, negative binomial-based outlier modeling, and

sketching with locality-sensitive hashing. The innovation of such methods—including RareQ—lies not in inventing a new mathematical index, but in its algorithmic operationalization to solve specific tasks. RareQ’s directed label propagation based on Q value is specifically tailed for identifying small and topologically self-organized subpopulations, which differs from conventional approaches based on uniform diffusion or modularity optimization (e.g., Louvain, Leiden). Below, we summarize RareQ’s key algorithmic innovations and ablation studies for benchmarking the contribution of its key components.

(i) Quantitative characterization of rare-cell neighborhood topology

RareQ distinguishes itself from local density estimation methods by characterizing the topological structure of the cell neighborhood graph; specifically, it quantifies the degree of interconnectivity among nodes within each cell’s k -nearest-neighbor graph. This enables identification of tightly connected cell cliques. Cells within smaller clusters exhibit higher Q values compared to those in larger populations, highlighting the effectiveness of the Q value in characterizing their relative topological isolation, as demonstrated in Figure 1b and Figure 5b of the manuscript.

(ii) Q-guided directed label propagation

RareQ employs a Q-guided label-propagation strategy that enables the detection of rare cell clusters. This includes not only topologically isolated groups but also, crucially, those embedded within or adjacent to major populations—a challenge for conventional graph-based or density-based frameworks where such rare cells are often absorbed into their abundant neighbors. RareQ overcomes this limitation by introducing a cell-specific Q metric that quantifies local topological coherence. During label propagation, cells with lower Q values adopt the label of the highest-Q neighbor within their local neighborhood. In this way, high-Q rare-cell cliques function as stable, topology-driven anchors that guide label diffusion outward, establishing a directional propagation process. This mechanism effectively captures both types of challenging rare populations. To assess the importance of this design, we performed ablation studies in which the Q-guided propagation was replaced by uniform diffusion (equivalent to assigning a constant $Q = 0$). This substitution resulted in decreased F1 scores, precision, and recall for several rare clusters (Fig. R1a – e), underscoring the essential role of Q-guided propagation in accurately identifying rare cell populations. We speculated that this decline specifically affects clusters that have weak separation from dominant populations, as those completely isolated within the KNN graph would be easily detected by conventional methods. To test our hypothesis, we created

detection difficulty through two distinct approaches: (1) we subsampled rare cells from major clusters and perturbed the expression of 10–100 genes by three-fold up regulation (Fig. R1f). (2) we adapted the real data shown in Supplementary Figure 23 and incorporated varying numbers of differentially expressed genes into the real rare clusters (Fig. R1g). The results showed that RareQ consistently outperformed the ablated version, confirming that Q-guided propagation enhances identification of challenging populations. Moreover, in practical applications such as the AD snRNA-seq dataset, several clusters became undetectable when the Q-values were removed (Fig. R2a, b).

Fig. R1. Ablation analyses of Q-value demonstrate its contribution to rare-cell identification across simulation and real datasets

- a. Cumulative curves comparing F1 scores before and after Q-value ablation in simulated datasets 1 and 2.
- b. Same as (a), but based on simulated dataset 3.
- c. Same as (a), but for NMI comparison in three simulated datasets.
- d. Comparison of F1 score, precision, and recall before and after Q-value ablation in real datasets.
- e. Same as (d), but for NMI comparison in the real datasets.
- f. Ablation analysis of the impact of Q-value under simulated conditions with varying degrees of gene-expression perturbations.
- g. Same as (f), but examining the impact of varying the number of differentially expressed genes in the simulation.

Fig. R2. Ablation analysis of the Q-value demonstrates its contribution to rare-cell identification in the Alzheimer's disease dataset

- a. UMAP plots showing the clusters before and after Q-value ablation in Alzheimer's disease (AD) snRNA-seq dataset.
- b. Confusion matrix comparing the identified cluster before and after Q ablation.

(iii) ΔQ_c -based topology-aware recursive merging

Following propagation, RareQ employs a recursive merging strategy driven by the cluster-level Q metric (Q_c). Merges are accepted only if the resulting cluster exhibits a higher Q_c value, reflecting increased internal connectivity. This provides a biologically interpretable, topology-based criterion that prevents over-clustering while preserving stable rare-cell clusters. Ablation analysis omitting this step led to a slightly increased F1 score but significantly decreased NMI, confirming that ΔQ_c -guided merging significantly improves overall global clustering accuracy, despite potential underclustering in a few clusters (Fig. R3).

Fig. R3. Ablation analysis of the merging component demonstrates its contribution to guarantee the clustering quality

- a. Cumulative curves comparing F_1 scores before and after merging step ablation in simulated datasets 1 and 2.
- b. Same as (a), but based on simulated dataset 3.
- c. Same as (a), but comparing NMI in ablation analysis across three simulated datasets.
- d. Cumulative curves comparing F_1 scores before and after merging ablation in 20 real scRNA-seq datasets.
- e. Same as (d), but comparing NMI.

In summary, RareQ's core methodological innovation is the extension of a static small-world descriptor into a topology-aware propagation and merging strategy. This approach significantly enhances the detection of hard-to-detect rare cells while maintaining the integrity of major clusters. The corresponding ablation analyses are presented in supplementary figure 65-66, and all scripts are openly accessible on GitHub (<https://github.com/xiaolab-xjtu/RareQ-reproducible-scripts>) to ensure reproducibility.

2. The definition is “low abundance and biological significance evidenced by distinct molecular profiles”. The supporting evidence largely relies on DEGs, GSEA, and signature scores, which are circular with respect to the same transcriptomic features used for discovery. No orthogonal validation (imaging, flow, perturbation, or independent

pathological assessment) is provided.

Response:

We appreciate the reviewer revisiting this critical concern and the opportunity for further clarification. We acknowledge the reviewer's concern that differential gene expression (DEG) analysis, gene set enrichment analysis (GSEA), and signature scoring might appear circular, as their purpose was to evaluate whether the identified rare populations exhibit coherent and biologically interpretable molecular programs.

We would like to reiterate that, for the Xenium mouse brain spatial transcriptomics dataset, we have already provided extensive orthogonal evidence in the previous revision to substantiate the rare CA3 subpopulations identified by RareQ. The validation data include in situ hybridization, RNAscope, riboprobe-based assays, and an independent scRNA-seq dataset. These data consistently confirmed the specificity and existence of the CA3 subpopulations specifically detected by RareQ. In this revision, we further validated our findings using two independent mouse brain spatial transcriptomics datasets from 10x Genomics Datasets (<https://www.10xgenomics.com/datasets>) (Fig. R4).

Fig. R4. Independent spatial transcriptomics datasets validate the spatial localization of CA3 subpopulations and their marker gene's expression.

a. Expression patterns of marker genes for CA2, CA3, and the two CA3 subtypes in the mouse brain spatial transcriptomics dataset.

b. Same as (a), showing the expression patterns in another mouse brain dataset.

Markers of CA3_1: Strip2; CA3_2: Rspo2, Plcx2, Nwd2.

To achieve the most rigorous validation possible, we have now incorporated several additional validation datasets, specifically:

(1) For the airway epithelial dataset, we leveraged three independent human spatial transcriptomics datasets (Murthy P. *et al.*, 2022, PMID: 35355018; Madissoon E. *et al.*, 2023, PMID: 36543915) and confirmed the presence of the RareQ-discovered Cycling Basal (C3), Ciliated (C4), and Secretory cell (C10) populations in specific regions along the airway lumen (Fig. R5), as defined by their RareQ-identified markers (Fig. 5d, e). These spatial datasets provide both molecular and contextual information that surpasses the limitations of bulk analysis (like flow cytometry) or purely morphological imaging approaches, thereby enabling significantly more precise validation.

(2) For the Alzheimer's disease (AD) dataset, we validated the significant enrichment of the C15 microglial population signature in AD patients compared with healthy controls using an independent bulk RNA-seq cohort (Fig. R6).

Fig. R6. Independent bulk RNA-seq confirms the significant enrichment of C15 microglial populations in Alzheimer's disease (AD) patients versus control individuals.

(3) In the B cell lymphoma datasets, the simultaneous scRNA-seq and scATAC-seq inherently provide cross-modality data validation. Furthermore, rare cell types, such as the PD-L1⁺ dendritic cells specifically identified by RareQ, have been independently reported in other published studies (Zilionis R *et al.*, 2019, PMID: 30979687; Zhang Q *et al.*, 2019, PMID: 31675496; Maier, B. *et al.*, 2020, PMID: 32269339).

Collectively, these multi-level validations provide independent, reproducible, and biologically grounded support for the rare populations detected by RareQ. To clarify, performing additional orthogonal experimental validations for every rare cell cluster would require specialized biological systems and substantial resources that fall beyond the scope of this computational methodology study. Nevertheless, our current findings establish a solid computational and biological foundation for future dedicated experimental investigations.

3. The added lymphoma, AD snRNA seq, and RCC spatial multi omics datasets are welcome. However, the analyses do not demonstrate control for donor level effects, batch, or case/control imbalance, particularly for the AD microglia where cluster 15 is "enriched in AD". What is the statistical framework used to test group specific enrichment after accounting for donor as a random effect, and potential covariates; otherwise the disease association claims are difficult to interpret.

Response: We thank the reviewer for this insightful comment. To account for donor-level effects, we performed differential cell abundance analysis using a negative binomial generalized linear model (NB-GLM) with quasi-likelihood (QL) estimation,

consistent with the approach implemented in the Milo algorithm (Dann E *et al.*, 2022, PMID: 34594043). In this model, donor identity was included as a random effect within a quasi-likelihood mixed model to account for intra-donor correlations and inter-donor variability. Group-specific enrichment (e.g., Alzheimer’s disease versus control) was then assessed using a QL F-test with specified contrasts for each cluster.

As shown in the Table R1 and Supplementary Table 6, among the three microglial subtypes (clusters 15, 16, and 21), cluster 15 exhibited a statistically significant enrichment of AD-derived cells after adjusting for donor effects, consistent with previously reported AD-associated microglial phenotypes. This statistical framework has now been explicitly detailed in the revised Methods section.

Table R1. Differential abundance of cell subtypes in AD vs. healthy controls

Cluster	Log2 FC	LogCPM	F-statistic	p value
1	-1.5403	10.1705	1.0636	0.3040
2	3.6042	10.0398	4.0582	0.0457
3	0.8819	10.6279	0.3307	0.5660
4	2.0888	11.4919	1.1927	0.2765
5	5.1107	11.5020	4.3424	0.0388
6	-0.0828	11.9391	0.0028	0.9573
7	-2.9078	12.1873	3.2537	0.0732
8	-1.5442	12.4684	1.0134	0.3156
9	0.0329	11.6586	0.0003	0.9848
10	-0.0552	12.4467	0.0014	0.9701
11	-3.4253	12.5718	4.1951	0.0422
12	1.0213	12.0423	0.4811	0.4889
13	-0.2559	13.0149	0.0393	0.8429
14	0.7780	12.6065	0.3025	0.5830
15	3.5935	13.1931	5.4319	0.0211
16	0.0236	13.1318	0.0003	0.9852

17	5.1205	13.4050	8.8075	0.0034
18	-2.2309	14.7938	2.1299	0.1465
19	-4.5116	14.8527	6.5940	0.0112
20	4.0969	14.7445	5.3698	0.0218
21	0.2677	15.7650	0.0533	0.8177
22	0.7794	14.9526	0.3262	0.5687
23	-0.5291	16.2058	0.2086	0.6484
24	-0.0865	17.4168	0.0055	0.9413
25	0.3220	19.3085	0.3615	0.5485

4. One of my important concerns was that RareQ's annotations on public spatial datasets did not match the original study reports. I do not see a systematic, side by side reconciliation with the original labels or a documented re annotation protocol.

Response: We thank the reviewer for raising this important point. Among the three public spatial datasets analyzed in our study, only the EEL FISH dataset included cell-type annotations in the original publication; the remaining datasets were released by the 10x Genomics platform without accompanying cell-type labels. As explained in our previous revision, for the EEL FISH dataset, we were unable to reproduce the original annotations even when re-clustering with more sensitive parameters following the given methods. Given this limitation, we have moved the results of this dataset to the Supplementary Figure 54.

In this revision, we conducted a systematic, side-by-side reconciliation of RareQ-derived annotations with the original cluster labels for the EEL FISH dataset, which is presented in the newly added Supplementary Figure 54d. This analysis revealed that the original annotations contained substantially more clusters than those identified by RareQ (Fig. R7a). We speculate that, if spatial context is not taken into account, the original annotations based solely on expression profiles of 440 genes may have been over-clustered, as demonstrated in our response to Comment 5. We also re-annotated this dataset following the procedures described in the original publication, using a resolution that produced a comparable number of clusters, and obtained similar results (Fig. R7b). To enhance transparency and reproducibility, we have provided a comprehensive re-annotation protocol on Github (<https://github.com/xiaolab-xjtu/RareQ-reproducible-scripts>).

Fig. R7. Confusion matrices comparing RareQ clusters with reference annotations.

- a. RareQ vs. original annotations.
- b. RareQ vs. Seurat re-annotations.

5. CHOIR is applied asymmetrically, with no equivalent CHOIR validation for high resolution Seurat baselines under comparable features and parameters. Moreover, because CHOIR builds RF classifiers on the same expression space used to define clusters, there is a risk of self validation.

Response: We have now conducted CHOIR analyses under comparable settings for both RareQ and the high-resolution Seurat baselines. In the Xenium spatial transcriptomics dataset, CHOIR indicated that most clusters identified by both RareQ and Seurat showed low probabilities of over-clustering (Fig. R8). However, for the EEL FISH spatial dataset, CHOIR evaluation suggested that the majority of clusters from the original annotations (116 of 188) and from Seurat re-clustering (73 of 188) were flagged as over-clustered (Fig. R9), only a minor subset of RareQ-derived clusters exhibited this issue (Supplementary figure 57a).

Fig. R8. Assessment of over-clustering using CHOIR for RareQ-derived clusters (a) and Seurat-derived clusters (b) on Xenium spatial transcriptomic data.

Fig. R9. Assessment of over-clustering using CHOIR for original clusters (a) and Seurat reannotated clusters (b) on EEL Fish spatial transcriptomic data.

We appreciate the reviewer's concern regarding potential self-validation within the CHOIR analysis, as the classifier operates on the same expression space used for

clustering. It's important to note that RareQ inherently mitigates over-clustering through its dedicated post hoc merging module. Furthermore, we used the Silhouette Coefficient—an effective internal metric for assessing cluster count appropriateness and overall quality—to independently evaluate RareQ's clustering quality against Seurat (Fig. R10a). This analysis showed that RareQ's clusters generally achieved a higher quality score than those generated by Seurat. For the separate EEL FISH dataset, we further demonstrated that clusters derived from both the original annotation and Seurat re-clustering were generally of low quality (Fig. R10b).

Fig. R10. Assessment of cluster quality using silhouette coefficients

a. Assessment of RareQ-derived clusters and Seurat-derived clusters on Xenium spatial transcriptomic data.

b. Assessment of original clusters and Seurat reannotated clusters on EEL Fish spatial transcriptomic data.

Furthermore, the RareQ's unique rare clusters showed distinct biological identities, evidenced by differential gene expression when compared to their most similar major clusters (Fig. R11). Finally, as detailed in our response to Comment 2, we conducted orthogonal validation of the RareQ-identified clusters. Across all analyses performed, RareQ consistently demonstrated that the resulting clusters are biologically meaningful and not artifacts of over-clustering.

Fig. R11. Heatmap illustrating the molecular differences between RareQ-specific rare cell clusters and their similar major cell types.

6. *Were MarsGT hyperparameters tuned comparably? Were identical kNN graphs/latent spaces supplied where applicable? Were input features harmonized (e.g., WNN vs. integrated embeddings)?*

Response: Following the original paper's methodology, we performed a grid search using four out of the ten multiome datasets to optimize MarsGT's hyperparameter

combination, including weighted decay, learning rate, and label smoothing. Our results indicated that MarsGT achieves its best performance with the settings: weighted decay (wd) = 0.3, learning rate (lr) = 0.0005, and label smoothing ($labsm$) = 0.3 (Fig. R12a). This set was subsequently adopted as the optimized parameter combination for all ten multiome datasets.

We further input the WNN-integrated features into MarsGT (denoted as MarsGT_WNN), matching the input used for RareQ. While MarsGT_WNN showed substantial improvements in rare-cell detection accuracy (F_1 score, precision, and recall) and overall clustering performance (NMI) compared to the default MarsGT input, its rare-cell detection accuracy remained slightly lower than that of RareQ_WNN (Fig. R12b). This new comparative analysis has been incorporated into the revised manuscript, and the results are discussed. Note that due to the long runtime of MarsGT, it is difficult to complete testing on large-scale simulated datasets. Therefore, we did not include MarsGT_WNN results on the simulation datasets.

Fig. R12. Hyperparameter tuning of MarsGT and performance comparison under its optimal parameter settings.

a. F₁ score comparison across different combinations of weight decay, learning rate, and label-smoothing parameters.

b. Performance comparison among MarsGT results obtained using the optimal parameter set and WNN-integrated embeddings as input (MarsGT_WNN) and those generated by RareQ.

7. In response to my request to avoid the umbrella term “spatial omics” if only RNA was used, you added one Xenium RNA+ADT dataset. This is helpful but still limited to a single sample and modality pair. Either temper the general claim (e.g., “spatial transcriptomics and one transcript protein dataset”) or broaden the evidence (additional tissues/samples or orthogonal spatial modalities).

Response: We thank the reviewer for this thoughtful comment. In the revised manuscript, we now describe the application of RareQ to spatial transcriptomics datasets and one transcript–protein (RNA+ADT) dataset, replacing the broader term “spatial omics.” This revision provides a more accurate and precise description of the modalities analyzed while maintaining clarity.

8. The tie break rule (smallest index) is deterministic but the biological stability of final clusters under index permutations is only assessed via aggregate metrics (F1/NMI).

Response: In response, we further evaluated clustering stability using the Jaccard index, a non-aggregate metric sensitive to local cell label differences. This analysis assessed the consistency of both cell memberships and cluster-specific gene signatures across runs. The results showed that random index selection produced outcomes nearly identical to those of the deterministic rule, with only minor variance observed in a few simulations (Fig. R13). To enhance flexibility, we have also implemented an optional non-deterministic tie-break mode, allowing users to perform multiple random runs and derive consensus clusters for obtaining higher-precision cluster assignments.

Fig. R13. Evaluating the impact of tie-break rules on RareQ stability

a. Effect of index permutation on the consistency of cell membership assignments in three simulation datasets.

b. Effect of index permutation on the stability of the top 50 highly specific marker genes per cluster in the tree simulation datasets.

c. similar to (a) but test on the 20 real datasets.

d. similar to (b) but test on the 20 real datasets.

9. In challenging spatial settings, RareQ may not add biological insight beyond recapitulating known structures.

Response: We thank the reviewer's concern. Spatial transcriptomics data are conceptually similar to scRNA-seq data but incorporate spatial information. In our study, RareQ was applied using expression data only, without incorporating spatial priors, making its accuracy in spatial transcriptomic analysis analogous to that in scRNA-seq. Therefore, we anticipate that RareQ can provide comparable biological insights in spatial contexts, as demonstrated in the Xenium mouse brain data, where RareQ uncovered CA3 subpopulations within known anatomical regions that competing methods failed to detect.

In this revision, we extend these findings by uncovering additional biological insights: In the EEL FISH mouse brain dataset, RareQ identified a rare choroid plexus epithelial subpopulation (C1) characterized by high expression of *Foxj1*, *Otx2*, and *Lratd2*, spatially nested within the dominant epithelial cluster (C11) (Fig. R14a-c). The spatial localization of the C1 population, along with its high expression of cilium-associated genes such as *Foxj1*, suggests that this group likely corresponds to ciliated epithelial cells of the choroid plexus. Previous studies have shown that cilia play a critical role in regulating cerebrospinal fluid flow and maintaining the ventricular microenvironment. Therefore, the rare ciliated-like choroid plexus epithelial population identified in our study provides a higher-resolution cellular reference for elucidating the fine structural organization and functional architecture of the ventricular system. Consistently, in another independent spatial dataset from an early developmental stage of the mouse brain, we observed a corresponding epithelial population marked by *Foxj1* and *Otx2* positioned adjacent to the main choroid plexus clusters defined by Ttr (Fig. R14d, e). We further validated these findings using an independent, high-resolution choroid plexus scRNA-seq dataset, where epithelial cells could be precisely mapped to the corresponding clusters in the EEL FISH spatial data (Fig. R14f, g). Of the 16 epithelial populations detected by RareQ, clusters like C3, C4, and C5 showed elevated expression of *Foxj1*, *Otx2*, and *Lratd2*, mirroring the characteristics of the spatially-defined C1 cluster (Fig. R14h). Together, these observations indicate that the RareQ identified rare cells in spatial context represent a biologically meaningful, specialized functional state, rather than technical noise.

Overall, RareQ not only recapitulates known spatial structures but also detects subtle and biologically meaningful rare-cell populations.

Fig. R14. Identification and characterization of choroid plexus cell subtypes.

- Original annotated spatial map of the choroid plexus population.
- Spatial distribution of the C1 and C11 choroid plexus populations identified by RareQ.
- Bubble plot showing marker genes of the C1 population.
- Spatial distribution of diverse cell types of a mouse head sagittal section.
- Expression patterns and spatial localization of the choroid plexus marker genes.
- RareQ-identified choroid plexus cell populations in mouse brain snRNA-seq data.
- Cross-modal correlation of epithelial cells (snRNA-seq) with all cells in the EEL FISH spatial dataset.
- Bubble plot illustrating marker genes for the c1 choroid plexus population within the snRNA-seq cell clusters.

10. To facilitate verification, please (i) include end to end scripts that reproduce a subset of the new disease analyses (e.g., AD microglia and RCC), (ii) pin package versions and random seeds, and (iii) provide exact commands for all figures cited in the response (R series).

Response: We have implemented four tutorials and supplied the requested end-to-end

scripts to guarantee complete reproducibility at GitHub repository (<https://github.com/xiaolab-xjtu/RareQ/tree/main/tutorials>; <https://github.com/xiaolab-xjtu/RareQ-reproducible-scripts>). Specifically, this repository includes:

- 1) Complete scripts for all new analyses, including the Alzheimer's disease (AD) microglia and renal cell carcinoma (RCC) case studies.
- 2) Pinned package versions and dependencies, with fixed random seeds utilized across all analyses.
- 3) A detailed guide containing the exact R commands required to reproduce every figure cited in our responses.

After carefully reviewing the point-by-point response and new analyses, I find that the core concerns remain unresolved. In its current form, the manuscript is not suitable for publication in Nature Communications.

Response: We appreciate the reviewer's continued engagement and careful evaluation. In this revision, we have thoroughly addressed all concerns through extensive new analyses, expanded validation datasets, and detailed methodological clarifications. We respectfully hope the reviewer will reconsider the revised manuscript in light of the new evidence and improvements provided.

Reviewer #2 (Remarks to the Author):

The authors have responded comprehensively to my comments.

Response: We are grateful for the reviewer's satisfaction with our responses and sincerely appreciate the time and effort invested in improving our manuscript.

The additional analyses would justify the choice of hyperparameters and should be included in the final manuscript.

Response: We appreciate the reviewer's input and have added these analyses to the revised manuscript. We detail the systematic evaluation of RareQ's sensitivity to hyperparameters, including the Q threshold, the Q ratio threshold for cluster merging, and the k parameter. The analyses confirm RareQ's remarkable stability across the parameter ranges tested and demonstrate that the default settings achieve optimal performance in most scenarios. The full evaluation is documented in the Supplementary Note 2.

Reviewer #2 (Remarks on code availability):

The Github repository has been updated to include the simulations. However, the quality of documentation around this and the tutorials is low and could be significantly improved to enable a competent person to reproduce the results in a reasonable amount of time. There are, for example, hard coded file paths and commented out lines in the simulation scripts. The effort that has been expended to enable a third party to read and understand the code is minimal.

Response: We thank the reviewer for emphasizing the importance of clear documentation and reproducibility. We have added four step-by-step R markdown tutorials—scRNA_analysis.Rmd, scRNA_ADT_analysis.Rmd, scRNA_scATAC_analysis.Rmd, and Xenium_spatial_analysis.Rmd—demonstrating the complete analysis workflow for single-cell RNA-seq, CITE-seq, paired scRNA-seq & scATAC-seq, and Xenium spatial datasets, respectively. In addition, all scripts have been standardized: hard-coded file paths were replaced with relative paths or clear directory instructions, and commented lines were cleaned and clarified to improve readability and transparency.

Reviewer #3 (Remarks to the Author):

My comments have been properly addressed. I recommend the publication of the manuscript. (Zixuan Cang)

Response: We are pleased that all comments have been properly addressed, and we appreciate your time and effort invested in reviewing our manuscript.

Reviewer #1 (Remarks to the Author):

This revised manuscript introduces RareQ, a graph-based clustering framework designed for rare-population discovery through Q-guided label propagation and Q_c-guided recursive merging. While the authors have addressed some previous comments, several fundamental methodological concerns remain unresolved. Most notably, the justification for the method's performance and hyperparameter selection relies heavily on supervised metrics, which contradicts the "unsupervised discovery" premise. Furthermore, the distinction between RareQ and existing graph-clustering pipelines remains insufficiently demonstrated.

Response: We thank the reviewer for the careful assessment of our revised manuscript. In this revision, we have (i) revised the hyperparameter discussion by removing potentially misleading statements and adding a label-free parameter selection/diagnostic framework, and (ii) expanded the analyses and comparisons to more explicitly demonstrate RareQ's conceptual and empirical distinction from standard graph-clustering approaches. Below, we address each concern in detail.

1. If RareQ is intended for unsupervised discovery, the reliance on label-dependent metrics (e.g., F1-score, NMI, Precision/Recall) to justify hyperparameter choices and "high-accuracy regions" is problematic. These metrics are unavailable in real-world discovery scenarios. Specifically, the suggestion in Supplementary Note 2 that users "modify parameters accordingly" to improve accuracy is impractical without ground-truth labels. The authors must provide a fully label-free parameter selection strategy and demonstrate its efficacy on benchmark datasets without invoking ground truth.

Response: Thank you for this insightful critique. Our original discussion in Supplementary Note 2 was intended to illustrate RareQ's robustness across a broad parameter range; however, we acknowledge that the phrase "modify parameters accordingly" could be interpreted as requiring access to ground-truth labels, which are typically unavailable in real-world discovery settings. We have therefore removed this statement and implemented a fully label-free parameter selection framework that does not rely on external annotations. This framework is guided by three internal diagnostic criteria:

- (1) **Rare-cluster credibility (Q):** assesses local connectivity and internal cohesiveness.
- (2) **Cluster independence (Q_c):** measures cluster-level separability by comparing intra-cluster connectivity to total connectivity.
- (3) **Cross-parameter consensus (NMI_{stability}):** represents the median pairwise NMI between the clustering obtained at a given parameter setting and those obtained under all other settings in the tested parameter grid, providing a label-free consensus measure across the parameter landscape.

To help users navigate the trade-offs among these diagnostics, we provide a ternary plot (Fig. R1a) summarizing their joint behavior across parameter combinations. Overall, RareQ shows robust performance across a broad parameter range. We further highlight the high-performing region and recommend selecting parameters from the Pareto-optimal frontier—settings that jointly maximize cohesiveness, independence, and stability.

Importantly, parameters are chosen solely based on these label-free diagnostics. We then report F_1 /NMI only as a post hoc benchmarking to validate the effectiveness of our proposed strategy. As shown in Fig. R1b, parameter settings within the trade-off region yield better classification of rare cells and major cell populations compared with settings from other regions of the ternary plot.

Fig. R1. Label-free parameter selection strategy for RareQ based on ternary plots. (a) The first column presents ternary plots summarizing the relative contributions of three label-free diagnostics (Q , Q_c , and $NMI_{stability}$) across different parameter combinations. The second column shows zoomed-in ternary plots based on min–max scaling metrics. **(b)** Boxplots comparing rare-cell detection performance (F_1 score) and global clustering accuracy (NMI) between parameter settings located near the center of the ternary plot and those farther from the center.

2. *Given that only MarsGT appears to have undergone systematic tuning, while most baseline models run with default settings, could the authors provide a predefined and fully fair benchmarking protocol? Furthermore, tuning should be conducted under the same search space/stopping criteria using unlabeled rules. Label-based metrics (e.g., F1/NMI) should not be used for tuning.*

Response: We thank the reviewer for this comment. We first clarify that our benchmarking does not tune any method using ground-truth labels. All baselines, including MarsGT, were run with author-recommended default settings as described in the original publications and/or official documentation. This is a widely accepted benchmarking practice because it evaluates each method under configurations intended and validated by its developers rather than under arbitrarily chosen parameterizations. Ground-truth annotations are used only for post hoc evaluation, which is standard for benchmarking clustering and rare-cell discovery methods. To ensure full reproducibility and transparency, we had provided a predefined benchmarking protocol (introduced in the previous revision): identical preprocessing and input graphs where applicable, together with scripts and configuration files.

Regarding the request to tune all methods under the same search space and stopping criteria, we respectfully note that a uniform search budget is not well-defined and not necessarily fair across heterogeneous algorithms. Their hyperparameters govern fundamentally different mechanisms (e.g., graph resolution for Leiden/Louvain, sketching and outlier scoring for FiRE, Gini cutoffs for GiniClust2, outlier thresholds for RaceID, and training dynamics for learning-based models), and their computational costs differ substantially. Imposing an identical search space/stopping rule can therefore bias the comparison and, in many cases, would replicate extensive sensitivity analyses already reported in the original method papers.

That said, to directly address the reviewer’s concern, we performed label-free parameters tuning using a predefined search grid and identical stopping criteria across methods (as similar to Fig. R1a). Parameter selection was guided only by our label-free diagnostics (i.e., internal cohesiveness/credibility, cluster independence, and cross-parameter consensus), with F_1/NMI reported strictly as retrospective evaluation. As shown in Fig. R2a, even under this standardized, label-free tuning protocol—allowing each baseline to select its best trade-off region—RareQ consistently achieves the strongest performance, consistent with our main conclusions.

In addition, we conducted a complementary sanity-check analysis to ensure our conclusions do not depend on any particular “best-case” configuration. Specifically, within each method’s author-recommended parameter range, we evaluated multiple representative parameter combinations (i.e., sampling diverse settings to span the plausible configuration space) and compared performance across methods.

As shown in Fig. R2b&c, RareQ again consistently achieves the best results across these diverse parameter settings.

Together, both the requested label-free selection experiment and the broader parameter-range sanity check independently support the same conclusion: RareQ's performance advantage is robust and not driven by asymmetric tuning or a single favorable parameter choice.

Fig. R2. Benchmarking RareQ against existing rare-population detection methods using a label-free parameter selection strategy across 20 scRNA-seq datasets. (a) Comparison of rare-cell detection performance (F_1 score, precision, and recall) and global clustering accuracy (NMI) for the indicated methods, with parameter combinations selected using the ternary plot-based, label-free strategy. **(b)** Boxplots summarizing rare-cell detection performance across multiple sampled parameter combinations for each method. Each dot represents the result for a given method and parameter setting, with metric values aggregated across all rare cell types in the 20 datasets. **(c)** Same as (b), but showing global clustering accuracy measured by NMI.

3. Regarding the EEL FISH dataset: The current explanation attributing discrepancies to "over-clustering" is insufficient. A concrete, step-by-step reconciliation analysis is required, including parameter sweeps and an identification of which preprocessing or resolution choices drive the divergence from original annotations.

Response: We thank the reviewer for this comment. Upon re-examining the original EEL FISH study, we realized that our previous interpretation of the reported '187 clusters' was inaccurate. We now understand that these clusters were derived from a cross-modal integration process (using Bonefight to map external scRNA-seq references, PMID: 36138169), which is conceptually distinct from a purely de novo

clustering approach. Consequently, we withdraw the statement regarding over-clustering in the EEL study.

With this corrected understanding, it is clear that the previously perceived 'granularity discrepancy' does not stem from preprocessing or resolution choices, but from a fundamental difference in data analytical objectives. While the original study leveraged external reference data to achieve high-resolution mapping, our analysis demonstrates that RareQ can perform autonomous de novo discovery using only the measured 440-gene EEL FISH slice.

Therefore, we focus on the fact that—even without an external reference—RareQ robustly recovers key hippocampal anatomical structures. This highlights RareQ's ability to extract biologically meaningful spatial organization and identify rare-cell anchors from a targeted gene panel alone.

4. The framework currently appears to be an operationalization of classical graph statistics (clustering coefficients) rather than a fundamentally new algorithmic contribution. The authors should explicitly formalize the mathematical or conceptual novelty that distinguishes RareQ from deterministic propagation. Please provide a direct comparison against strong, generic baselines (e.g., standard label propagation variants, or Leiden/Louvain with resolution control) using the exact same kNN graph. This is necessary to prove that RareQ's performance cannot be replicated by simple tuning of existing, widely-used pipelines.

Response: As explained in our manuscript, the novelty of RareQ lies in characterizing a distinctive topological signature of rare populations on the single-cell kNN graph. Specifically, rare cells often form locally cohesive, clique-like microstructures with reduced external connectivity compared with major cell populations (Fig. 1b). RareQ operationalizes this topology into an explicit inference mechanism for rare-population discovery. To quantify this signature, we introduced a cell-wise neighborhood connectivity metric, Q .

Conceptually, Q is motivated by the same intuition as the clustering coefficient in small-world networks, i.e., measuring the “cliquishness” of a local neighborhood. However, Q is not a direct reuse of the classical clustering coefficient; rather, it is formulated to reflect the topology typically observed in single-cell kNN graphs, where rare populations are frequently embedded within dominant groups. Specifically, the classical clustering coefficient mainly measures how densely a node's neighbors connect to each other (local compactness). In contrast, Q captures not only within-neighborhood cohesion but also the relative separation of that neighborhood from its surroundings: a cell attains a higher Q value when its neighbors are densely interconnected while exhibiting comparatively fewer connections to cells outside the neighborhood. This enables Q to jointly characterize local cohesion and relative independence (boundary sharpness), which is

particularly informative for deeply embedded rare populations. Because the standard clustering coefficient is defined on undirected graphs, directly substituting it for Q on directed kNN graphs would be mathematically inconsistent. Q can be viewed as a kNN-graph-compatible reformulation of neighborhood “cliquishness” used by RareQ as an operational signal in inference rather than a post hoc statistic.

Importantly, in RareQ, Q is not used merely as a passive descriptive statistic. Instead, it is embedded into the inference procedure. During Q -guided label propagation, high- Q local cliques act as stable topological anchors, directing label diffusion along highly consistent neighborhood structures until convergence. This propagation dynamics is mechanistically distinct from standard majority-vote label propagation or deterministic propagation variants. We further introduce a Q_c -guided recursive merging strategy to recover appropriate granularity for major populations without sacrificing rare-population separability, thereby preventing global over-clustering.

To directly address the reviewer’s question of whether RareQ can be replicated by simple tuning of widely used pipelines, we performed a same-graph benchmark against a strong generic baseline: Seurat’s default Louvain clustering. Using exactly the same kNN graph, we ran RareQ with default parameters and swept Louvain’s resolution across a broad range (0.5, 0.8, 1, 2, 3, 4, 5, 6, 7, 8, 9, 10, 15, 20), evaluating rare-population detection (F_1 , precision, recall) and global clustering consistency (NMI). We note that resolutions as high as 15–20 are far beyond typical Seurat usage and are included here solely as a stress test to examine whether aggressive tuning can reproduce RareQ’s behavior.

The results show that Louvain approaches RareQ’s rare-cell detection performance only at extremely high resolutions (15–20) (Fig. R3a). However, these settings induce pronounced over-clustering, excessively fragmenting major populations and substantially reducing global clustering consistency (markedly lower NMI) (Fig. R3a, b). In contrast, RareQ achieves both strong rare-population recovery and high global coherence without requiring extreme parameter settings. These results indicate that, while modularity-based clustering may theoretically approximate rare-population recovery through aggressive resolution tuning, it does so at the cost of generating many over-partitioned subclusters—making it difficult in practice to reliably distinguish true rare populations from artifacts of over-splitting in an unsupervised setting.

In summary, we emphasize that RareQ’s methodological novelty does not lie in proposing an entirely new notion of “neighborhood compactness,” but in:

- (1) Formalizing a cell-wise topological measure (Q) tailored to single-cell kNN graphs to capture rare-population cohesion and boundary sharpness, rather than directly adopting classical clustering coefficients;
- (2) Operationalizing this topology into a Q -guided, topology-aware label propagation mechanism in which high- Q cliques serve as stable anchors for diffusion and convergence; and

(3) Coupling propagation with a Q_c -guided recursive merging strategy to suppress global over-clustering while preserving rare-population separability.

Consequently, the characteristic behavior of RareQ cannot be reproduced by simple parameter tuning of existing modularity-based clustering methods.

Fig. R3. Performance comparison between RareQ and Louvain clustering. (a) Comparison of RareQ and Seurat’s default Louvain clustering on identical kNN graphs. RareQ was applied using default parameters, while Louvain clustering was evaluated across a range of resolution values (0.5–20). **(b)** The number of detected rare cell clusters by RareQ and Louvain clustering.

5. Rare population discovery is highly sensitive to technical noise. How robust is RareQ to upstream choices (e.g., number of PCs, HVG selection, batch correction methods, or k)? A systematic analysis is needed to show if/when RareQ produces spurious clusters due to density gradients or batch effects. The authors should include "negative control" experiments (e.g., permuted data, batch-only simulations, or datasets without true rare types) to quantify the False Positive Rate (FPR) of the method. Practical

diagnostics for users to identify potential failure modes in unlabeled settings should also be provided.

Response: We thank the reviewer for this comment. In response, we conducted a systematic robustness evaluation across real datasets and designed dedicated negative-control simulations to quantify false positives. We also provide practical, label-free diagnostics to help users identify potential failure modes in unlabeled settings.

(1) Robustness to upstream choices on real scRNA-seq datasets

To assess sensitivity to upstream analytical choices, we evaluated RareQ on 20 real scRNA-seq datasets while varying key preprocessing settings, including the number of PCs, HVG selection, and batch-correction methods. Overall, we observed that RareQ remains stable and achieves consistently strong performance when using sufficiently informative feature spaces, specifically when PCs ≥ 40 and HVGs $\geq 1,500$ (Fig. R4a, b). Across these settings, RareQ's rare-population detection and global clustering performance showed minimal variation, indicating robust behavior under commonly used preprocessing regimes.

For the k parameter (denoted as `k.param` to distinguish from RareQ's neighborhood size k parameter) in kNN graph construction, we previously performed a dedicated sensitivity analysis in the first round of revision (Supplementary Fig. 24) by evaluating `k.param` in the range of 10–30. RareQ exhibited remarkable robustness across this range, with consistently stable rare-cell detection and overall clustering performance. Together with the simulation results presented in the manuscript and the additional analyses reported in Response 1, these evaluations support that RareQ is robust to typical parameter variations and maintains strong tolerance to technical noise.

(2) Batch-effect stress test

To explicitly test whether RareQ may spuriously identify “rare populations” driven purely by batch composition, we designed a controlled simulation using the Splatter package (PMID: 28899397). We simulated datasets split into two batches and two biological cell types, with $\sim 1,500$ cells in total. We constructed a challenging scenario in which one batch within cell type 1 constitutes only 1% of all cells, whereas the remaining batch of the same cell type and both batches of cell type 2 each comprise $\sim 33\%$ of cells (Fig. R4c). We further varied the batch-effect strength by increasing Splatter's `batch.facLoc` parameter from 0.1 to 1.0.

We then tested RareQ under (i) no batch correction and (ii) several commonly used batch-correction strategies, including CCA, RPCA, SCT, and Harmony (Fig. R4d). We asked whether RareQ would incorrectly call the rare batch (the 1% subset) as a rare “cell population”. Without batch correction, RareQ indeed identified the rare batch subset as a rare population, yielding FPR = 1 in this constructed batch-only rarity setting. With batch correction, RareQ largely avoided this failure mode when the

batch effect was mild to moderate, and did not spuriously call the rare batch subset as a rare population (Fig. R4e).

Notably, when the batch effect became extremely strong, batch correction may over-collapse biological distinctions such that batches across cell types become confounded, which can lead to mis-grouping. Among the tested approaches, CCA and SCT integration showed the most favorable behavior in preserving biological structure while mitigating batch-only artifacts. Overall, these results support a clear practical conclusion: when batch effects are present, applying batch correction is critical to prevent spurious “rare populations” that are driven primarily by batch composition rather than biology.

(3) Negative-control dataset without true rare populations and FPR estimation

To further quantify the false positive rate under a “no true rare types” scenario, we constructed a negative-control dataset by extracting low-heterogeneity major populations from the UUO kidney dataset. This control dataset is designed to lack genuine rare populations. Across a range of PCs and HVG settings, RareQ produced near-zero FPR, indicating that RareQ does not tend to invent rare clusters in relatively homogeneous data when true rare populations are absent (Fig. R4f).

(4) Practical diagnostics for unlabeled settings

Finally, we provide practical diagnostics that users can apply when ground-truth labels are unavailable. First, because RareQ is topology-driven, the distribution of Q values offers an immediate label-free sanity check. In the UUO-kidney-derived negative-control dataset, Q -value distributions are similar across clusters, consistent with the absence of a topologically distinct rare clique (Fig. R4g). In contrast, when we spiked in a known rare population (e.g., JGA cells), the rare population exhibits substantially elevated Q values compared to other clusters, consistent with the intended topological signature that RareQ leverages (Fig. R4h).

Second, users can further assess biological plausibility using standard downstream analyses, such as differential expression/marker discovery (e.g., FindMarkers), to evaluate whether the detected rare cluster shows coherent marker programs consistent with a plausible biological state rather than technical artifacts.

We have incorporated these robustness analyses, negative controls, and user-facing diagnostics into the revised Supplementary information (see Supplementary Notes 4, 6, and 7), addressing the reviewer’s concerns about technical confounding and failure-mode detection in unlabeled settings.

Fig. R4. Robustness to upstream choices and batch-effect correction. (a-b) Robustness of RareQ to principle components (PCs) (a) and highly variable genes (HVGs) choices across real scRNA-seq datasets (b). (c) A simulation data with a batch-only rare cell cluster generated by the Splatter package. (d) False positive rate (FPR) assessment of RareQ on simulated data before batch correction and after batch correction with different methods across increasing batch-effect strengths; batch-cofounded clusters are highlighted. (e) Cell populations identified by RareQ without batch correction and after batch correction using the CCA-based integration. (f) A negative-control dataset constructed from low-heterogeneity major cell populations extracted from the UUO kidney dataset; across a range of PCs and HVG settings, RareQ maintains near-zero FPR. (g) In the UUO-kidney-derived negative-control dataset, Q -value distributions are comparable across clusters. (h) Q -value-based diagnostic after spiking in a true rare population (JGA cells).

Reviewer #1 (Remarks to the Author)

The authors have provided additional analyses and clarifications in response to my previous review. Nevertheless, several key methodological issues remain only partially addressed.

1. The authors withdraw the earlier “over-clustering” statement and attribute the discrepancy to differing analysis goals. This, however, does not address my original request for a step-by-step reconciliation. Please clarify what is treated as the “ground truth”/annotation for EEL FISH in your evaluation, how the mapping/alignment between outputs and annotations is performed, and provide parameter sweeps that demonstrate what specifically drives agreement versus disagreement.

Response:

In our previous revision, we removed the statement about “over-clustering” and clarified that the observed differences in cluster number arise from fundamentally different analytical goals between the two approaches. To directly address the reviewer’s remaining concerns:

(1) Ground truth / reference annotation used in our evaluation: we treat the cell-type annotations reported in the original EEL FISH study (the 187 clusters) as the reference “ground truth” for evaluation purposes. These annotations were generated by the original authors via cross-modal reference mapping, integrating the EEL FISH spatial data with an external single-cell transcriptomic atlas. We use them as the external reference to assess whether RareQ, when run in a purely de novo manner on the 440-gene spatial transcriptomics data alone, is able to recover the rare-cell populations identified in the original study—particularly in the hippocampus region.

(2) Mapping / alignment between RareQ clusters and reference annotations: we do not attempt a one-to-one cluster-to-cluster reconciliation because the two clustering procedures are methodologically distinct (cross-modal reference mapping vs. unsupervised clustering directly on spatial data). Instead, we perform a standard many-to-many comparison using a confusion matrix between RareQ’s cluster assignments and the original study’s cell-type annotations (Supplementary Figure 54d). To quantify the overall agreement between the two partitions, we compute NMI. This metric is reported in the revised manuscript and serves as our primary measure of alignment with the reference annotations.

(3) Why a step-by-step reconciliation is not feasible: because the original 187 clusters were not derived by clustering the EEL FISH data itself but by transferring labels from a separate single-cell RNA-seq reference, the two clusterings reflect different sources of information and different resolution objectives. A forced direct reconciliation would therefore be biologically and methodologically inappropriate. Our goal is not to reproduce the exact cross-modal reference mapping result, but to evaluate whether RareQ can discover biologically meaningful rare populations when given only the spatial transcriptomics data.

(4) Regarding the request for additional parameter sweeps

We have already conducted extensive parameter sensitivity analyses across multiple datasets (including different resolutions, k values, and other hyperparameters) in previous rounds of

review. These analyses are summarized in the Supplementary Note 4. Given that the core discrepancy is explained by the difference in input data and methodological strategy, we do not expect further parameter sweeps on this particular dataset to meaningfully alter or resolve the observed differences in cluster number. Accordingly, we have not added additional sweeps for this specific comparison in the revision.

In addition, during the second round of revision, we addressed a similar concern by clustering the EEL FISH data directly using Seurat and performing a head-to-head comparison with RareQ's clustering results.

2. Same-graph comparisons using Louvain resolution sweeps are informative, but they are not sufficient to support the claim that the approach cannot be reproduced by standard graph pipelines. The authors should include same graph comparisons with Leiden (resolution sweeps) and standard label propagation/LPA variants, with clearly specified stopping rules. In addition, key ablations (e.g., removing Q-guidance, removing Qc-based merging, or replacing Q with a closely related statistic) are needed to demonstrate which components are necessary for the reported gains.

Response:

To address the reviewer's concern, we have expanded the baseline comparisons and revisited the ablation evidence.

(1) Same-graph comparisons with Leiden (implemented in Seurat) resolution sweeps

We have added comprehensive Leiden clustering results obtained on the same kNN graphs used by RareQ (Fig. R1). The results reveal a consistent pattern across datasets: Leiden can approach RareQ's rare-cell detection performance (e.g. F_1 score) only when using very high-resolution parameters. However, such aggressive resolution settings cause severe over-partitioning (dramatically increased number of clusters), which is not practically usable in an unsupervised setting for two main reasons: there is no dataset-agnostic and robust stopping rule for selecting an optimal resolution, and excessive splitting makes it very difficult to reliably distinguish true rare subpopulations from over-splitting artifacts without external ground-truth labels. In contrast, RareQ achieves strong rare-cell recovery performance without requiring such extreme hyperparameter tuning.

(2) Label propagation (LPA) on the same graphs

We further evaluated the standard label propagation algorithm (LPA) using the widely adopted `cluster_label_prop` function from the `igraph` package (v1.3.0), applied to the identical kNN graphs across all 20 scRNA-seq benchmark datasets. LPA was run with its default convergence criterion. While LPA achieves competitive F_1 scores in some datasets, it consistently underperforms RareQ in terms of Normalized Mutual Information (NMI). This indicates that LPA is less effective at producing globally coherent partitions on the same graph structure. Together with the Louvain and Leiden results, these findings reinforce that RareQ's advantage is not simply explained by applying standard community detection methods to the same input graph.

(3) Ablation studies to identify drivers of the reported gains

The key ablations requested (removing Q -guidance, removing Q_c -based merging, etc.) were already performed and reported in detail during the second round of revision (Supplementary Note 1). These experiments show that:

- i. Removing Q -guidance leads to clear drops in rare-cell detection performance (F_1 score) across the majority of datasets;
- ii. Omitting the Q_c -based merging step results in a modest increase in F_1 in some cases, but causes a significant decrease in NMI, confirming the importance of this component for maintaining globally coherent and biologically meaningful partitions.

These results provide direct evidence that the performance gains of RareQ are specifically attributable to the proposed Q -guided propagation and Q_c -constrained quality-controlled merging mechanisms, rather than to generic improvements in graph partitioning.

Fig. R1 Benchmarking RareQ against Seurat (Leiden algorithm) with different resolutions in 20 scRNA-seq datasets. **a.** Boxplots comparing RareQ's F_1 score and NMI with Seurat (Leiden algorithm) with different resolutions. **b.** Number of original cell types and clusters inferred by RareQ and Seurat (Leiden algorithm) in (a).

Fig. R2 Benchmarking RareQ against label propagation algorithm (LPA) in 20 scRNA-seq datasets.
a. Boxplots comparing RareQ's F_1 score and NMI with LPA from igraph R package. **b.** Number of original cell types and clusters inferred by RareQ and LPA in (a).

(Remarks on code availability)

The code can be further optimized, and the description can also be more detailed.

Response: We thank the reviewer for this invaluable suggestion. We will further optimize the code and refine the documentation in the upcoming version.